# Variational Inference on the Final-Layer Output of Neural Networks

**Yadi Wei**  *weiyadi@iu.edu*
*Luddy School of Informatics, Computing, and Engineering*
*Indiana University*

**Roni Khardon**  *rkhardon@iu.edu*
*Luddy School of Informatics, Computing, and Engineering*
*Indiana University*

**Reviewed on OpenReview:** *https://openreview.net/forum?id=mTOzXLmLKr*

## Abstract

Traditional neural networks are simple to train but they typically produce overconfident predictions. In contrast, Bayesian neural networks provide good uncertainty quantification but optimizing them is time consuming due to the large parameter space. This paper proposes to combine the advantages of both approaches by performing Variational Inference in the Final layer Output space (VIFO), because the output space is much smaller than the parameter space. We use neural networks to learn the mean and the variance of the probabilistic output. Using the Bayesian formulation we incorporate collapsed variational inference with VIFO which significantly improves the performance in practice. On the other hand, like standard, non-Bayesian models, VIFO enjoys simple training and one can use Rademacher complexity to provide risk bounds for the model. Experiments show that VIFO provides a good tradeoff in terms of run time and uncertainty quantification, especially for out of distribution data.

## 1 Introduction

With the development of training and representation methods for deep learning, models using neural networks provide excellent predictions. However, such models fall behind in terms of uncertainty quantification and their predictions are often overconfident (Guo et al., 2017). Bayesian methods provide a methodology for uncertainty quantification by placing a prior over parameters and computing a posterior given observed data, but the computation required for such methods is often infeasible. Variational inference (VI) is one of the most popular approaches for approximating the Bayesian outcome, e.g., (Blundell et al., 2015; Graves, 2011; Wu et al., 2019). By minimizing the KL divergence between the variational distribution and the true posterior and constructing an evidence lower bound (ELBO), one can find the best approximation to the intractable posterior. However, when applied to deep learning, VI requires sampling to compute the ELBO, and it suffers from both high computational cost and large variance in gradient estimation. Wu et al. (2019) have proposed a deterministic variational inference (DVI) approach to alleviate the latter problem. The idea relies on the central limit theorem, which implies that with sufficiently many hidden neurons, the distribution of the output of each layer forms a multivariate Gaussian distribution. Thus we only need to compute the mean and covariance of the output of each layer. However, DVI still suffers from high computational cost and complex optimization.

Inspired by DVI, we observe that the only aspect that affects the prediction is the distribution of the output of the final layer in the neural network. We therefore propose to perform variational inference in the final-layer *output space* (rather than parameter space), where the posterior mean and diagonal variance are learned by a neural network. We call this method VIFO. Like all Bayesian methods, VIFO induces a distribution over

its probabilistic predictions and has the advantage of uncertainty quantification in predictions. At the same time, VIFO has a single set of parameters and thus enjoys simple optimization as in non-Bayesian methods.

We can motivate VIFO from several theoretical perspectives. First, we derive improved priors (or regularizers) for VIFO motivated by collapsed variational inference (Tomczak et al., 2021) and empirical bayes (Wu et al., 2019). The new regularizers greatly improve the performance of VIFO. Second, we show that, for the linear case, with expressive priors VIFO can capture the same predictions as standard variational inference. On the other hand, with practical priors and deep networks VIFO exhibits limited expressiveness. We propose to overcome this limitation by using ensembles that enable fast training and further improve uncertainty quantification. Third, due to its simplicity, one can derive risk bounds for the model through Rademacher complexity. Thus, VIFO was motivated as an effective simplification of VI and DVI, but the ensembles of VIFO can be seen as a Bayesian extension of Deep Ensembles (Lakshminarayanan et al., 2017). We discuss the connections to other Bayesian predictors below.

An experimental evaluation compares VIFO with VI and other state of the art approximation methods and with non-Bayesian neural networks. The results show that (1) VIFO is much faster than VI and only slightly slower than base models, and (2) ensembles of VIFO achieve better uncertainty quantification on shifted and out-of-distribution data while preserving the quality of in-distribution predictions. Overall, VIFO provides a good tradeoff in terms of run time and uncertainty quantification especially for out-of-distribution data.

## 2 VIFO

In this section we describe our VIFO method in detail. We start with a description of the non-Bayesian *base model*. Given a neural network parametrized by weights $W$ and input $x$, the output is $z = f_W(x) \in \mathbb{R}^K$. The base model provides probabilistic predictions by combining the output of the network with any prediction likelihood $p(y|z)$. Traditional, non-Bayesian models, minimize $-\log p(y|z)$ or a regularized variant.

*Remark* 2.1. The base model and VIFO are applicable with any likelihood function and our development of VIFO below is general. To illustrate we discuss classification and regression. In classification, $K$ is the number of classes. The probability of being class $i$ is defined as

$$p(y = i|z) = \text{softmax}(z)_i = \frac{\exp z_i}{\sum_j \exp z_j}. \tag{1}$$

In regression, $z = (m, l)$ is a 2-dimensional vector and $K = 2$. We apply a function $g$ on $l$ that maps $l$ to a positive real number. The probability of the output $y$ is:

$$p(y|z) = \mathcal{N}(y|m, g(l)) = \frac{1}{\sqrt{2\pi g(l)}} \exp\left(-\frac{(y-m)^2}{2g(l)}\right). \tag{2}$$

By fixing the weights $W$, base models map $x$ to $z$ deterministically. Bayesian inference puts a distribution over $W$ and marginalizes out to get a distribution over $z$ from which predictions can be calculated. Since exact marginalization is not tractable, variational inference provides an approximation which yields the well known ELBO objective for optimization:

$$\log p(\mathcal{D}) \geq \mathbb{E}_{q(W)}\left[\log \frac{p(W, \mathcal{D})}{q(W)}\right] = \sum_{(x,y)\in\mathcal{D}} \mathbb{E}_{q(W)}[\log p(y|W, x)] - \text{KL}(q(W)\|p(W)). \tag{3}$$

As shown by Wu et al. (2019), by the central limit theorem, with a sufficiently wide neural network the marginal distribution of $z$ is Gaussian. DVI explicitly calculates an analytic approximation of the mean and variance of the output of each layer (valid for specific activation functions) and avoids the sampling typically used for optimization of the ELBO in other methods.

VIFO pursues this in a direct manner. It has two sets of weights, $W_1$ and $W_2$ (with potentially shared components), to model the mean and variance of $z$. That is, $\mu_q(x) = f_{W_1}(x)$, $\sigma_q(x) = g(f_{W_2}(x))$, where $g : \mathbb{R} \to \mathbb{R}^+$ maps the output to positive real numbers as the variance is positive. Thus, $q(z|x) = \mathcal{N}(z|\mu_q(x), \text{diag}(\sigma_q^2(x)))$, where $\mu_q(x), \sigma_q^2(x)$ are vectors of the corresponding dimension. We will call $q(z|x)$ the *variational output distribution*. As in the base model, given $z$, $y$ is generated from the likelihood $p(y|z)$.

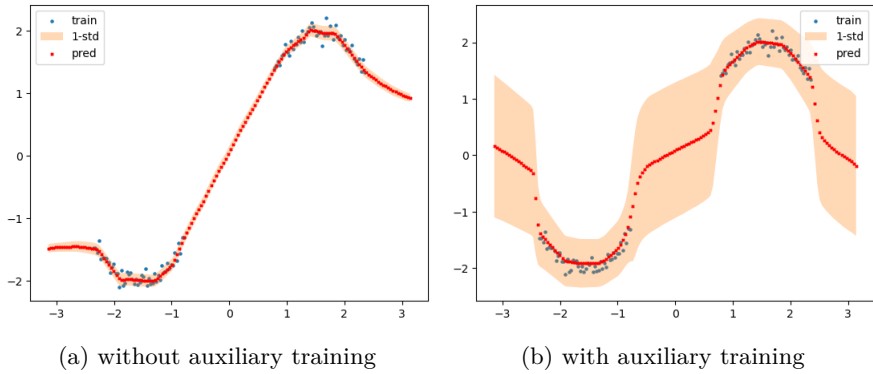

(a) without auxiliary training          (b) with auxiliary training

Figure 1: Predictive distribution of VIFO using an MLP. Blue points are training data generated from a sinusoidal function, red points are the predicted mean, shaded area indicates the 1 standard deviation. More details are in Appendix D.1.

*Remark* 2.2. VIFO in regression is different from the existing models known as the mean-variance estimator (Kabir et al., 2018; Khosravi et al., 2011; Kendall & Gal, 2017). Instead, mean-variance estimators are the base models that VIFO can be applied on. Applying VIFO to these models results in four outputs: $\mu_m$ and $\mu_l$, which are the means of $m$ and $l$, and $\sigma_m^2$ and $\sigma_l^2$, which are the variances of $m$ and $l$. These variances come from the variational output distribution. We sample $m \sim \mathcal{N}(\mu_m, \sigma_m^2)$ and $l \sim \mathcal{N}(\mu_l, \sigma_l^2)$, then form $z = (m, l)$. Like all Bayesian methods VIFO computes a distribution over distributions which is lacking in non-Bayesian predictions.

Unlike VI which puts a prior over $W$, VIFO models the distribution over $z$ and therefore we put a prior directly over $z$. We consider two options, a conditional prior $p(z|x)$ and a simpler prior $p(z)$. Both of these choices yield a valid ELBO using the same steps:

$$\log p(y|x) \geq \mathbb{E}_{q(z|x)}\left[\log \frac{p(y,z|x)}{q(z|x)}\right] = \mathbb{E}_{q(z|x)}[\log p(y|z)] - \mathrm{KL}(q(z|x)\|p(z|x)). \tag{4}$$

The approach has some similarity to Dirichlet-based models (Sensoy et al., 2018; Charpentier et al., 2020; Bengs et al., 2022). However, we perform inference on the output whereas, as discussed by Bengs et al. (2022), these models implicitly perform variational inference on the prediction. In particular, in that work $z$ is interpreted as a vector in the simplex and $q(z|x)$ and $p(z)$ are Dirichlet distributions, whereas when using VIFO for classification $z$ has a Gaussian distribution and $p(y|z)$ is on the simplex. In other words, we model and regularize different distributions. We discuss related work in more details below.

Eq. (4) is defined for every $(x, y)$. For a dataset $\mathcal{D} = \{(x, y)\}$, we optimize $W_1$ and $W_2$ such that

$$\sum_{(x,y)\in\mathcal{D}} \left\{ \mathbb{E}_{q(z|x)}[\log p(y|z)] - \mathrm{KL}(q(z|x)\|p(z|x)) \right\}$$

is maximized. We regard the negation of the first term $\mathbb{E}_{q(z|x)}[-\log p(y|z)]$ as the loss term and treat $\mathrm{KL}(q(z|x)\|p(z|x))$ as a regularizer.

## 2.1 Auxiliary Training

As in prior work (Sun et al., 2019), to improve uncertainty quantification we introduce auxiliary input $x_{\mathrm{aux}}$ and include $\mathrm{KL}(q(z|x_{\mathrm{aux}})\|p(z|x_{\mathrm{aux}}))$ as an additional regularization term. We include corresponding coefficients $\eta$ and $\eta_{\mathrm{aux}}$ on the regularizers, as is often done in variational approximations (e.g., (Higgins et al., 2017; Sheth & Khardon, 2017; Jankowiak et al., 2020; Wenzel et al., 2020; Wei et al., 2021; Wei & Khardon, 2022).). Then, viewed as a regularized loss minimization, the optimization objective for VIFO becomes:

$$\min_{W_1, W_2} \sum_{(x,y)\in\mathcal{D}} \left\{ \mathbb{E}_{q(z|x)}[-\log p(y|z)] + \eta \mathrm{KL}(q(z|x)\|p(z|x)) + \eta_{\mathrm{aux}} \sum_{x_{\mathrm{aux}}} \mathrm{KL}(q(z|x_{\mathrm{aux}})\|p(z|x_{\mathrm{aux}})) \right\}. \tag{5}$$

Generally the loss term is intractable, so we use Monte Carlo samples to approximate it. In practice, since auxiliary data is not available, we uniformly sample $x_{\text{aux}}^{(i)} \sim \text{Unif}[x_{\text{min}}^{(i)} - \frac{d}{2}, x_{\text{max}}^{(i)} + \frac{d}{2}]$ where $d = x_{\text{max}}^{(i)} - x_{\text{min}}^{(i)}$ for each entry $i$. Figure 1 shows an example where an MLP is used to learn a complex function over 1 dimensional input space, illustrating that such regularization can improve uncertainty quantification in the area where the data is missing.

## 2.2 Collapsed VIFO

Bayesian methods are often sensitive to the choice of prior parameters. To overcome this, Wu et al. (2019) used empirical Bayes (EB) to select the value of the prior parameters, and Tomczak et al. (2021) proposed collapsed variational inference, which defined a hierarchical model and performed inference on the prior parameters as well. We show how these ideas are applicable in VIFO and derive empirical Bayes as a special case of collapsed variational inference. In addition to $z$, we model the prior mean $\mu_p$ and variance $\sigma_p^2$ as Bayesian parameters. Now the prior becomes $p(z|\mu_p, \sigma_p^2)p(\mu_p, \sigma_p^2)$ and the variational distribution is $q(z|x)q(\mu_p, \sigma_p^2)$. Then the objective becomes:

$$
\log p(y|x) \geq \mathbb{E}_{q(z|x)q(\mu_p, \sigma_p^2)} \left[ \log \frac{p(y, z, \mu_p, \sigma_p^2|x)}{q(z|x)q(\mu_p, \sigma_p^2)} \right]
$$
$$
= \mathbb{E}_{q(z|x)}[\log p(y|z)] - \mathbb{E}_{q(\mu_p, \sigma_p^2)}[\text{KL}(q(z|x)\|p(z|\mu_p, \sigma_p^2))] - \text{KL}(q(\mu_p, \sigma_p^2)\|p(\mu_p, \sigma_p^2)). \quad (6)
$$

Similar to Eq. (5), we treat the first term as a loss and the other two terms as a regularizer along with a coefficient $\eta$ and aggregate over all data. Since the loss does not contain $\mu_p$ and $\sigma_p^2$, we can get the optimal $q^*(\mu_p, \sigma_p^2)$ by optimizing the regularizer and the choice of $\eta$ will not affect $q^*(\mu_p, \sigma_p^2)$. Then we can plug in the value of $q^*$ into Eq. (6). We next show how to compute $q^*(\mu_p, \sigma_p^2)$ and the final collapsed variational inference objective. The derivations are similar to the ones by Tomczak et al. (2021) but they are applied on $z$ not on $W$. Recall that $K$ is the dimension of $z$.

**Learn mean, fix variance** Let $p(z|\mu_p) = \mathcal{N}(z|\mu_p, \gamma I)$, $p(\mu_p) = \mathcal{N}(\mu_p|0, \alpha I)$. Then $q^*(\mu_p|x)$ is

$$
\underset{q(\mu_p)}{\arg\min} \, \mathbb{E}_{q(\mu_p)}[\text{KL}(q(z|x)\|p(z|\mu_p))] + \text{KL}(q(\mu_p)\|p(\mu_p)),
$$

and the optimal $q^*(\mu_p|x)$ can be computed as:

$$
\log q^*(\mu_p|x) \propto -\frac{(\mu_q(x) - \mu_p)^\top (\mu_q(x) - \mu_p)}{2\gamma} - \frac{\mu_p^\top \mu_p}{2\alpha},
$$

and $q^*(\mu_p|x) = \mathcal{N}(\mu_p|\frac{\alpha}{\alpha+\gamma}\mu_q(x), \frac{\alpha\gamma}{\alpha+\gamma})$. Notice that, unlike the prior, $q^*(\mu_p)$ depends on $x$. If we put $q^*$ back in the regularizer of Eq. (6), the regularizer becomes:

$$
\frac{1}{2\gamma}\left[1^\top \sigma_q^2(x) + \frac{\gamma}{\gamma+\alpha}\mu_q(x)^\top \mu_q(x)\right] - \frac{1}{2}1^\top \log \sigma_q^2(x) + \frac{K}{2}\log(\gamma+\alpha) - \frac{K}{2}. \quad (7)
$$

As in Tomczak et al. (2021), Eq. (7) puts a factor $\frac{\gamma}{\gamma+\alpha} < 1$ in front of $\mu_q(x)^\top \mu_q(x)$, which weakens the regularization on $\mu_q(x)$. We refer to this method as "VIFO-mean".

Figure 2, shows the learned prior for VIFO-mean and VI for the same example as in Figure 1. We observe that VIFO-mean allows for diverse prior distributions and captures the data distribution well.

**Other Regularizers** The same approach can be used for a joint prior $p(z|\mu_p, \sigma_p^2) = \mathcal{N}(z|\mu_p, \sigma_p^2)$, $p(\mu_p) = \mathcal{N}(\mu_p|0, \frac{1}{t}\sigma_p^2)$, $p(\sigma_p^2) = \mathcal{IG}(\sigma_p^2|\alpha, \beta)$, where $\mathcal{IG}$ is inverse Gamma, yielding a method we call "VIFO-mv". Similarly, the hierarchical prior in empirical Bayes models the variance but not the mean $p(\sigma_p^2) = \mathcal{IG}(\sigma_p^2|\alpha, \beta)$, $p(z|\sigma_p^2) = \mathcal{N}(z|0, \sigma_p^2)$ and yields "VIFO-eb". Derivations are given in Appendix B.

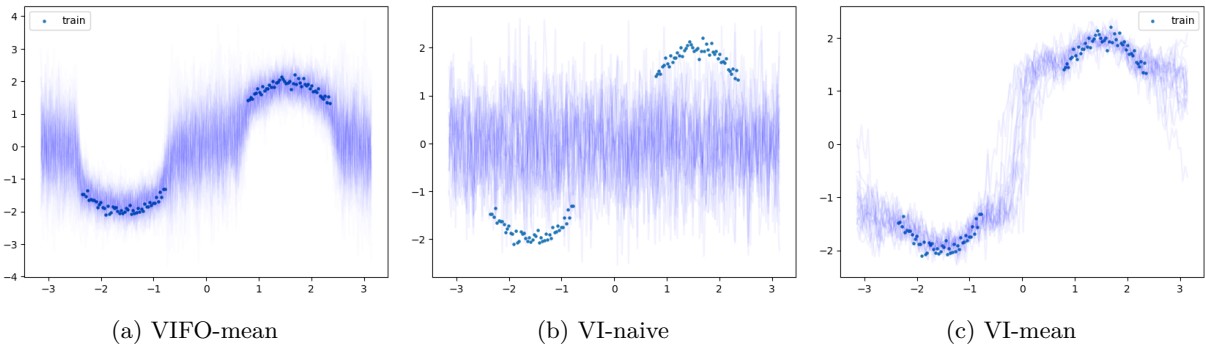

(a) VIFO-mean  (b) VI-naive  (c) VI-mean

Figure 2: Induced predictions by learned prior distribution for different methods. Note that VI has a prior over weights and VIFO has a prior over $z$. For each method we sample values from the prior and calculate predictions $y$ based on the sampled values. We then plot the $y$ values. As we can see, VI-naive induces a uniform prior that does not capture the data distribution, VI-mean has an increased variance in areas where data is missing and VIFO-mean does so to a larger extent. Details are given in Appendix D.1.

## 3 Expressiveness of VIFO

VIFO is inspired by DVI and it highly reduces the computational cost. In this section we explore whether VIFO can produce exactly the same predictive distribution as VI. We show that this is the case for linear models but that for deep models VIFO is less powerful. We first introduce the setting of linear models. Let the parameter be $\theta$, then the model is:

$$y|x, \theta \sim p(y|\theta^\top x). \tag{8}$$

For example, $p(y|\theta^\top x) = \mathcal{N}(y|\theta^\top x, \frac{1}{\beta})$ where $\beta$ is a constant for Bayesian linear regression; and $p(y = 1|\theta^\top x) = \frac{1}{1+\exp(-\theta^\top x)}$ for Bayesian binary classification.

For simplicity, we assume $\theta \in \mathbb{R}^d$, where $d$ is the dimension of $x$, and then the output dimension $K = 1$. The standard approach specifies the prior of $\theta$ to be $p(\theta) = \mathcal{N}(\theta|m_0, S_0)$, and uses $q(\theta) = \mathcal{N}(\theta|m, S)$. Then the ELBO objective, with a dataset $X_N = (x_1, x_2, \ldots, x_N) \in \mathbb{R}^{d \times N}$ and $Y_N = (y_1, y_2, \ldots, y_N) \in \mathbb{R}^N$, is

$$\sum_{i=1}^N \mathbb{E}_{q(\theta)}[\log p(y_i|\theta^\top x_i)] - \mathrm{KL}(q(\theta)\|p(\theta))$$

$$= \sum_{i=1}^N \mathbb{E}_{q(\theta)}[\log p(y_i|\theta^\top x_i)] - \frac{1}{2}[\mathrm{tr}(S_0^{-1}S) - \log |S_0^{-1}S|] - \frac{1}{2}(m - m_0)^\top S_0^{-1}(m - m_0) + \frac{d}{2}. \tag{9}$$

As the following theorem shows, if we use a conditional correlated prior and a variational posterior that correlates data points, then in the linear case VIFO can recover the ELBO and VI solution. We defer the proof and discussion of $K > 1$ to Appendix A.1.

**Theorem 3.1.** *Let $q(z|x) = \mathcal{N}(z|w^\top x, x^\top V x)$ be the variational predictive distribution of VIFO, where $w$ and $V$ are the parameters to be optimized, and let $p(z|X_N) = \mathcal{N}(z|m_0^\top X_N, X_N^\top S_0 X_N)$ and $q(z|X_N) = \mathcal{N}(z|w^\top X_N, X_N^\top V X_N)$ be a correlated and data-specific prior and posterior (which means that for different data $x$, we have a different prior/posterior over $z$). Then the VIFO objective is equivalent to the ELBO objective implying identical predictive distributions.*

However, as the next theorem shows, for the non-linear case we cannot produce the variational output distribution $q(z|x)$ as if it is marginalized over the posterior on $W$.

**Theorem 3.2.** *Given a neural network $f_W$ parametrized by $W$ and a mean-field Gaussian distribution $q(W)$ over $W$, there may not exist a set of parameters $\tilde{W}$ such that for all input $x$ we have $\mathbb{E}_{q(W)}[f_W(x)] = f_{\tilde{W}}(x)$.*

The proof is given in Appendix A.1. The significance of these results is twofold. On the one hand, we see from Theorem 3.2 and the conditions of Theorem 3.1 that the representation is more limited, i.e., efficiency comes at some cost. On the other hand, Theorem 3.1 shows the connection of VIFO to VI, which gives a better perspective on the approximation it provides. Moreover, this facilitates the use of existing improvements in VI for VIFO such as collapsed VI applied to VIFO.

In practice, a correlated and data-specific prior $p(z|x)$ is complex, and tuning its hyperparameters would be challenging. Hence, for a practical algorithm we propose to use a simple prior $p(z)$ independent of $x$. In addition, to reduce computational complexity, we do not learn a full covariance matrix and focus on the diagonal approximation. These aspects limit expressive power but enable fast training of VIFO and hence also ensembles of VIFO.

## 4 Rademacher Complexity of VIFO

In this section we provide generalization bounds for VIFO through Rademacher Complexity. We need to make the following assumptions. These assumptions hold for classification and with a smoothed loss for regression as shown in Appendix A.2.

**Assumption 4.1.** $\log p(y|z)$ is $L_0$-Lipschitz in $z$, i.e., $|\log p(y|z) - \log p(y|z')| \leq L_0\|z - z'\|_2$.

**Assumption 4.2.** The link function $g$ is $L_1$-Lipschitz.

Recall that the Rademacher complexity of a set of vectors $A \subseteq \mathbb{R}^N$ is defined as $R(A) = \frac{1}{N}E_{\sigma\sim\{-1,1\}^N}[\sup_{a\in A}\sum_i \sigma_i a_i]$. The Rademacher complexity of the set of loss values induced by functions $f \in \mathcal{F}$ over a dataset $S$ has been used to derive generalization bounds for learning of the class $\mathcal{F}$. We need the following technical lemma, proved in Appendix A.2, that generalizes well known Lipschitz based bounds (Shalev-Shwartz & Ben-David, 2014) to multi-input functions.

**Lemma 4.3.** *Consider an $L$-Lipschitz function $\phi : \mathbb{R} \times \mathbb{R} \to \mathbb{R}$, i.e. $\phi(a_1, b_1) - \phi(a_2, b_2) \leq L(|a_1 - a_2| + |b_1 - b_2|)$. For $\boldsymbol{a}, \boldsymbol{b} \in \mathbb{R}^N$, let $\phi(\boldsymbol{a}, \boldsymbol{b})$ denote the vector $(\phi(a_1, b_1), \dots, \phi(a_N, b_N))$. Let $\phi(A \times B)$ denote $\{\phi(\boldsymbol{a}, \boldsymbol{b}) : \boldsymbol{a} \in A, \boldsymbol{b} \in B\}$, then*

$$R(\phi(A \times B)) \leq L(R(A) + R(B)). \tag{10}$$

Applying the previous lemma sequentially over multiple dimensions we obtain:

**Corollary 4.4.** *Consider an $L$-Lipschitz function $\phi : \mathbb{R}^d \to \mathbb{R}$, i.e., for any $x, x' \in \mathbb{R}^d$, $\phi(x) - \phi(x') \leq L\|x - x'\|_1$. Let $\phi(A^d) = \{\phi(a_{1:d,i}) : \boldsymbol{a}_1, \boldsymbol{a}_2, \dots, \boldsymbol{a}_d \in A \subset \mathbb{R}^N\}$, then $R(\phi(A^d)) \leq LdR(A)$.*

With the assumptions and technical lemma, we derive the main result:

**Theorem 4.5.** *Let $\mathcal{H}$ be the set of functions that can be represented with neural networks with parameter space $\mathcal{W}$, $\mathcal{H} = \{f_W(\cdot)|W \in \mathcal{W}\}$. VIFO has two components, so the VIFO hypothesis class is $\mathcal{H} \times \mathcal{H} = \{(f_{W_1}(\cdot), f_{W_2}(\cdot)) \,|W = (W_1, W_2), W_1, W_2 \in \mathcal{W}\}$. Let $l$ be the loss function for VIFO, $l(W, (x, y)) = E_{q_W(z|x)}[-\log p(y|z)]$. Then the Rademacher complexity of VIFO is bounded as $R(l \circ (\mathcal{H} \times \mathcal{H}) \circ S) \leq 2(L_0 \max\{1, L_1\}K) \cdot R(\mathcal{H} \circ S)$, where $K$ is the dimension of $z$ and $S$ is training dataset.*

The proof is in Appendix A.2 and it shows how reparametrization can facilitate computation of Rademacher bounds for Bayesian predictors. The Rademacher complexity for VIFO is bounded through the Rademacher complexity of deterministic neural networks. This shows one advantage of VIFO which is more amenable to analysis than standard VI due to its simplicity. Risk bounds for VI have been recently developed (e.g., (Germain et al., 2016; Sheth & Khardon, 2017)) but they require different proof techniques. The Rademacher complexity for neural networks is $O\left(\frac{B_W B_x}{\sqrt{N}}\right)$ (Golowich et al., 2018), where $B_W$ bounds the norm of the weights and $B_x$ bounds the input. The Rademacher complexity of VIFO is of the same order.

## 5 Related Work

VIFO is related to but distinct from a number of variational and one-pass methods. Dirichlet-based methods (Sensoy et al., 2018; Charpentier et al., 2020; Bengs et al., 2022), discussed above, implicitly perform vari-

ational inference on the prediction and the network output provides parameters of a Dirichlet distribution. Like VIFO they provide Bayesian predictions in a single pass over the network, but their relation to the standard variational inference in parameter space is non obvious. On the other hand, VIFO is a single pass method clearly related to VI in parameter space which enables the benefits of collapsed variational inference. Thus VIFO can be seen to bridge between Dirichlet methods and VI. DUQ (van Amersfoort et al., 2020) provides an alternative approach using one pass on the network. It first embeds examples into a latent space, similar to $z$, but computes classification prediction and uncertainty quantification through RBF distances to centroids of classes in that space. Hence its predictions are very different. Another related line of work (Sun et al., 2019; Tran et al., 2022) performs variational inference in function space. However, they focus on choosing a better prior in weight space which is induced from Gaussian Process priors on function space, whereas VIFO directly induces a simple prior on function space. Sharma et al. (2023) model the distribution of the last layer by adding random noise as input and do not give an explicit form of the output distribution.

VIFO differs from other existing variational inference methods as well. The local reparametrization trick (Tomczak et al., 2020; Oleksiienko et al., 2022) reduces the variance from sampling in VI. This is done by performing two forward passes with the mean and variance at each layer before sampling the output for the layer. Hence this modifies the sampling process of VI whereas VIFO only requires one pass on the network and samples only the output of the last layer for prediction.

Last-layer variational inference (Brosse et al., 2020; Kristiadi et al., 2020; Daxberger et al., 2021; Liu et al., 2020; Harrison et al., 2024) performs variational inference on the *parameters* of the last layer, while we perform variational inference on the *output* of the last layer. Note that the last layer usually contains more parameters than the output which has constant size. Last Layer Laplace Kristiadi et al. (2020); Daxberger et al. (2021) reduces training complexity by first estimating the MAP solution, and then estimating the covariance of the parameters in one pass. SNGP (Liu et al., 2020) is a variant of this method, that aims to mimic the sensitivity of Gaussian processes to distances among examples, by incorporating Fourier features at the one to last layer. Finally, VBLL (Harrison et al., 2024) still maintains a distribution on last layer parameters, but approximates the expectation over these parameters in closed form (for specific likelihoods) to reduce the complexity of training and prediction. Thus, all these methods are much closer to VI because they maintain distributions over weights whereas VIFO produces distributions in output space.

VIFO shares some aspects with the model of Kendall & Gal (2017), where both use neural networks to output the mean and covariance of the last layer. However Kendall & Gal (2017) use the cross entropy loss, $-\log \mathbb{E}_{q(z|x)} p(y|z)$ instead of our loss in Eq. (5), they use dropout for epistemic uncertainty, and their objective has no explicit regularization. Hence unlike VIFO their formulation does not correspond to a standard ELBO. It is also interesting to compare VIFO to the Deep Variational information bottleneck (Alemi et al., 2017). The model is motivated from a different perspective but its final optimization objective, obtained after some approximations, is similar to our Eq. (4). In this sense the model is close to the VIFO-naive. However, in their formulation, $z$ is the output of a bottleneck layer which is not the final layer (because it is meant to constrain the information that flows to the final layer), and $p(z)$ which is the prior in our model is a posterior on the marginal posterior on $z$. Nonetheless, our development of rich priors through collapsed inference can help inform the choice of $p(z)$ in that model, which is typically taken to be a standard Normal.

Various alternative Bayesian techniques have been proposed. One direction is to get samples from the true posterior, as in Markov chain Monte Carlo methods Wenzel et al. (2020); Izmailov et al. (2021). Expectation propagation aims to minimize the reverse KL divergence to the true posterior (Teh et al., 2015; Li et al., 2015). These Bayesian methods, including variational inference, often suffer from high computational cost and therefore hybrid methods were proposed. Stochastic weight averaging Gaussian (Maddox et al., 2019) forms a Gaussian distribution over parameters from the stochastic gradient descent trajectory in the base model. Dropout (Gal & Ghahramani, 2016) randomly sets weights 0 to capture uncertainty in the model. Deep ensembles (Lakshminarayanan et al., 2017) use ensembles of base models learned with random initialization and shuffling of data points and then average the predictions. These methods implicitly perform approximate inference. In addition to these methods, there are also non-Bayesian methods to calibrate overconfident predictions, for example, temperature scaling (Guo et al., 2017) introduces a temperature parameter to anneal the predictive distribution to avoid high confidence. VIFO strikes a balance between simplicity and modelling power to enable simple training and Bayesian uncertainty quantification. On the one hand, VIFO

can be seen as a simplification of VI. On the other hand, it can be seen as an extension of the base model. From this perspective, the use of ensembles of VIFO, which extend the ensembles of Lakshminarayanan et al. (2017), are highly motivated as a practical algorithm.

## 6 Experiments

In this section, we compare the empirical performance of VIFO with VI and hybrid methods that use the base model. In VIFO, $W_1$ and $W_2$ share all parameters except those in the last layer. VI candidates include the VI algorithm ("VI-naive"(Blundell et al., 2015)) with fixed prior parameters, and other variations from collapsed variational inference (Tomczak et al., 2021) and empirical Bayes (Wu et al., 2019). Non-Bayesian and hybrid methods include the base model ("SGD", because it uses stochastic gradient descent as optimizer), stochastic weight averaging ("SWA", which uses the average of the SGD trajectory on the base model as the final weights) from Izmailov et al. (2018) and SWA-Gaussian ("SWAG", which uses the SGD trajectory to form a Gaussian distribution over the neural network weight space) from Maddox et al. (2019). We use ensembles of the base models which are known as deep ensembles (Lakshminarayanan et al., 2017), and the ensembles of SWAG models, which are the multiSWAG model of Wilson & Izmailov (2020), both of which are considered strong baselines for uncertainty quantification (Ovadia et al., 2019). In addition to these methods, we include other approximate Bayesian algorithms for comparison. These include repulsive ensembles ("Repulsive", (D'Angelo & Fortuin, 2021)), the Dirichlet-based model ("Dir", (Sensoy et al., 2018)), dropout ((Gal & Ghahramani, 2016)), last layer Laplace with prior optimization ("Laplace", (Daxberger et al., 2021)), and variational Bayes last layer ("VBLL", (Harrison et al., 2024)). Our main goal is to show:

- VIFO is much faster than VI and only slightly slower than base models;

- Ensembles of VIFO preserve the quality of in-distribution predictions;

- Ensembles of VIFO achieve better uncertainty quantification on shifted and out-of-distribution (OOD) data than all baselines.

For our main experiments, we pick four large datasets, CIFAR10, CIFAR100, SVHN, STL10, together with two types of neural networks, AlexNet (Krizhevsky et al., 2012) and PreResNet20 (He et al., 2016). The regularization parameter $\eta$ is fixed to 0.1 for both VIFO and VI, as this choice yields better performance compared with the standard choice $\eta = 1$. Empirically we observe that using collapsed variational inference in VI does not improve the performance. This is because Tomczak et al. (2021) used $\eta = 1$ to obtain their results whereas we use $\eta = 0.1$ which provides a much stronger baseline. For auxiliary training, we experiment with $\eta_{\text{aux}} \in \{0.0, 0.1, 0.5, 1.0\}$. Larger values of $\eta_{\text{aux}}$ generally improve OOD data detection at the cost of increased in-distribution loss, and there is no generic optimal value of $\eta_{\text{aux}}$. In our main paper, we present only the case where $\eta_{\text{aux}} = 0.1$ because it provides a balance between in-distribution and OOD performance, with performance of other choices of $\eta_{\text{aux}}$ provided in the appendix. In addition, VIFO-mean and VIFO-mv perform better than other variants of VIFO. Thus, we only list these variants in our main paper and provide full results for other variants for VIFO and VI in the appendix. For each method we run 5 independent runs and report means and standard deviations in results. Complete details for the setup and hyperparameters are given in Appendix D.2. Our code is available on `https://github.com/weiyadi/VIFO`.

### 6.1 Run Time

Ignoring the data preprocessing time, we compare the run time of training 1 epoch of VI, VIFO and the base model. In Table 1 we show the mean and standard deviation of 10 runs of these methods. Different regularizers do not affect run time, so we only show that of VI-naive for VI and VIFO-mean for VIFO. In addition, as shown in Table 1, VIFO is much faster than VI and is slightly slower than the base model. As shown in Fig. E.7, VIFO converges faster than, or as fast as VI. Consequently, the training time until convergence for VIFO is shorter than for VI.

Table 1: Running time (seconds) for training 1 epoch with batch size 512, AlexNet

| dataset | CIFAR10 | CIFAR100 | SVHN | STL10 |
|---------|---------|----------|------|-------|
| size | 50000 | 50000 | 73257 | 500 |
| VI | $8.51 \pm 0.41$ | $8.27 \pm 0.40$ | $11.56 \pm 0.39$ | $1.75 \pm 0.41$ |
| VIFO | $2.18 \pm 0.39$ | $2.17 \pm 0.43$ | $2.72 \pm 0.38$ | $1.16 \pm 0.40$ |
| base | $1.97 \pm 0.41$ | $1.99 \pm 0.43$ | $2.46 \pm 0.40$ | $1.12 \pm 0.38$ |

Table 2: Test log loss ($\downarrow$) of single VIFO and ensembles of VIFO.

|  | VIFO-mean | | VIFO-mv | |
|--|-----------|--|---------|--|
|  | single | ensemble | single | ensemble |
| CIFAR10 | $0.527 \pm 0.015$ | $0.345 \pm 0.003$ | $0.626 \pm 0.010$ | $0.324 \pm 0.001$ |
| CIFAR100 | $2.253 \pm 0.032$ | $1.688 \pm 0.006$ | $2.688 \pm 0.029$ | $1.725 \pm 0.003$ |
| STL10 | $1.333 \pm 0.065$ | $1.055 \pm 0.008$ | $1.531 \pm 0.019$ | $1.123 \pm 0.008$ |
| SVHN | $0.509 \pm 0.029$ | $0.351 \pm 0.005$ | $0.520 \pm 0.027$ | $0.298 \pm 0.009$ |

The differences in run time are dominated by sampling and forward passes in the network. Let $P$ denote the number of parameters in the *base* model and thus each forward/backward pass takes $O(P)$ time. The time complexity for computing the loss for each output of the base model is $O(1)$. The base model only needs 1 forward pass without sampling and thus the time complexity is $O(P)$. VIFO needs 1 forward pass and $M$ samples to compute the loss so the time complexity is $O(P + M)$. VI needs $M$ samples of the parameter space and $M$ forward passes, thus the time complexity is $O(PM + M) = O(PM)$. The same facts apply for predictions on test data, where the advantage can be important for real time applications.

## 6.2 Ensembles of VIFO

Theorem 3.2 points out that the expressiveness of VIFO is limited. To overcome this, we use ensembles of VIFO, which independently train multiple VIFO models and average their predictions. Section 6.1 establishes fast training of VIFO, allowing us to train VIFO models simultaneously while still maintaining the running time advantage of VIFO. We investigate the impact of ensemble size on performance in Appendix E.5. While increasing the ensemble size enhances performance, the improvement diminishes once the size exceeds 5. Therefore, we choose an ensemble size of 5. Table 2 shows that with ensembles, VIFO with auxiliary training achieves much better log loss than when using a single model. The same holds without auxiliary training. This indicates that ensembles of VIFO are much more expressive than a single VIFO. In the following experiments, we use ensembles of VIFO. **For a fair comparison in the remainder of the paper, we use ensembles for all methods** except for VI (which is time-consuming) and repulsive ensembles (which are themselves ensembles).

## 6.3 In-distribution Performance

In this section we use log loss and accuracy to measure the performance for in-distribution data.

Fig. 3 and Fig. E.2 in Appendix compare main methods in terms of log loss. First, we observe that repulsive ensembles and the Dirichlet method have much worse log loss than all other methods and they tend to give underconfident predictions. Second, we observe that using auxiliary training slightly increases the log loss but the increase is negligible. Later we can see that auxiliary training improves the uncertainty quantification for out-of-distribution data. We observe that VIFO is competitive with all methods in terms of log loss, with relatively small differences between the top group of methods in each case. Fig. E.1 and Fig. E.3 show accuracy on test data in the same experiments, revealing that in many cases VIFO outperforms VI and it is competitive with all methods. Finally, there is no clear winner between VIFO-mean and VIFO-mv; VIFO-mv provides a small advantage overall but might be more sensitive as illustrated by the performance on CIFAR100 with PreResNet20.

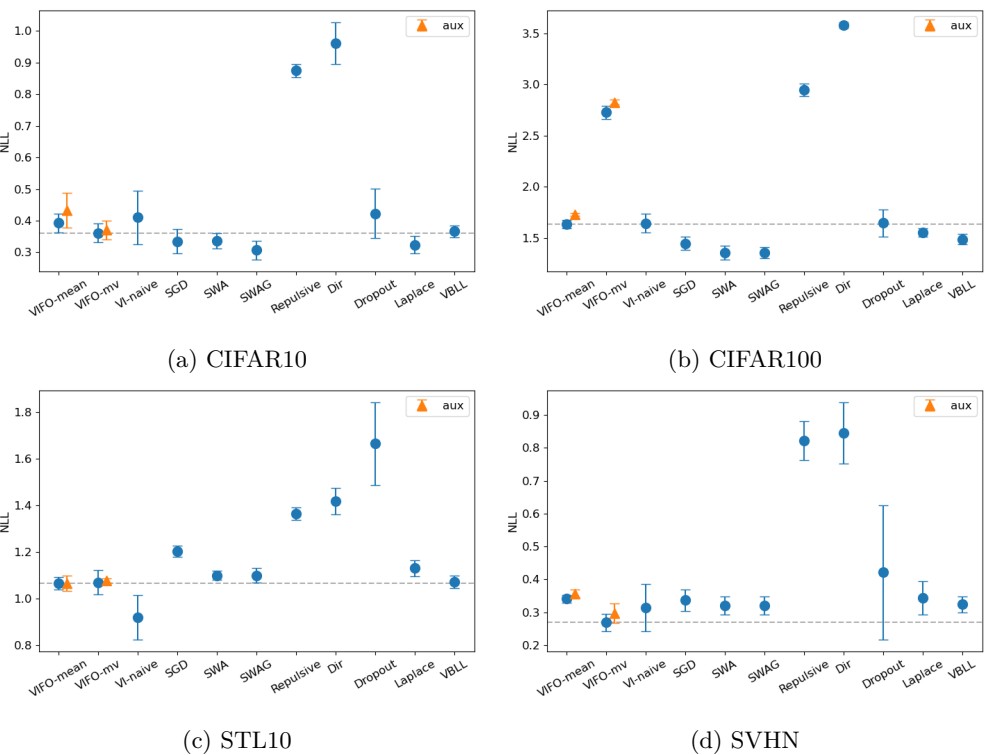

| (a) CIFAR10 | (b) CIFAR100 |
| --- | --- |

| (c) STL10 | (d) SVHN |
| --- | --- |

Figure 3: Test log loss (↓) on PreResNet20. Dashed lines indicate the best version of VIFO. The error bar is three times of the standard deviation for better visualization and same for other figures.

## 6.4 Uncertainty Quantification

In this section we examine whether VIFO can capture the uncertainty in predictions for shifted and OOD data. We measure performance using ECE, Entropy and AUC for detecting OOD data. These represent a comprehensive set of measures from the literature. For datasets, for uncertainty under data shift, STL10 and CIFAR10 can be treated as a shifted dataset for each other, as the figure size of STL10 is different from CIFAR10, and STL10 shares some classes with CIFAR10 so the labels are meaningful. For uncertainty under OOD data, we choose the SVHN dataset as an OOD dataset for CIFAR10 and STL10, as SVHN contains images of digits and the labels of SVHN are not meaningful in the context of CIFAR10.

**Expected Calibration Error (ECE)**  ECE (Naeini et al., 2015; Ovadia et al., 2019) is often used to measure the uncertainty quantification under data shift. We separate data into bins of the same size according to the confidence level, calculate the difference between the accuracy and the averaged confidence in each bin and then average the absolute differences among all bins. Better calibrated models have lower ECE. ECE has its faults (for example the trivial classifier has zero ECE) but it is nonetheless informative. We selected the number of bins to be 20.

Fig. 4 shows the ECE of each method under data shift. As we can see, both VIFO-mean and VIFO-mv achieve the best performance compared to all other methods.

**Entropy**  Entropy (Ovadia et al., 2019) of the categorical predictive distribution is used to measure the uncertainty quantification for out-of-distribution (OOD) data as the labels for OOD data are meaningless. We want our model to be as uncertain as possible and this implies high entropy and low confidence (the maximum probability assigned to any class) in the predictive distribution. We summarize the averaged entropy for the entire dataset in Fig. 5 and Fig. E.4. We can see that both VIFO-mean and VIFO-mv are better than all other methods except repulsive ensembles and the Dirichlet method. However, as observed

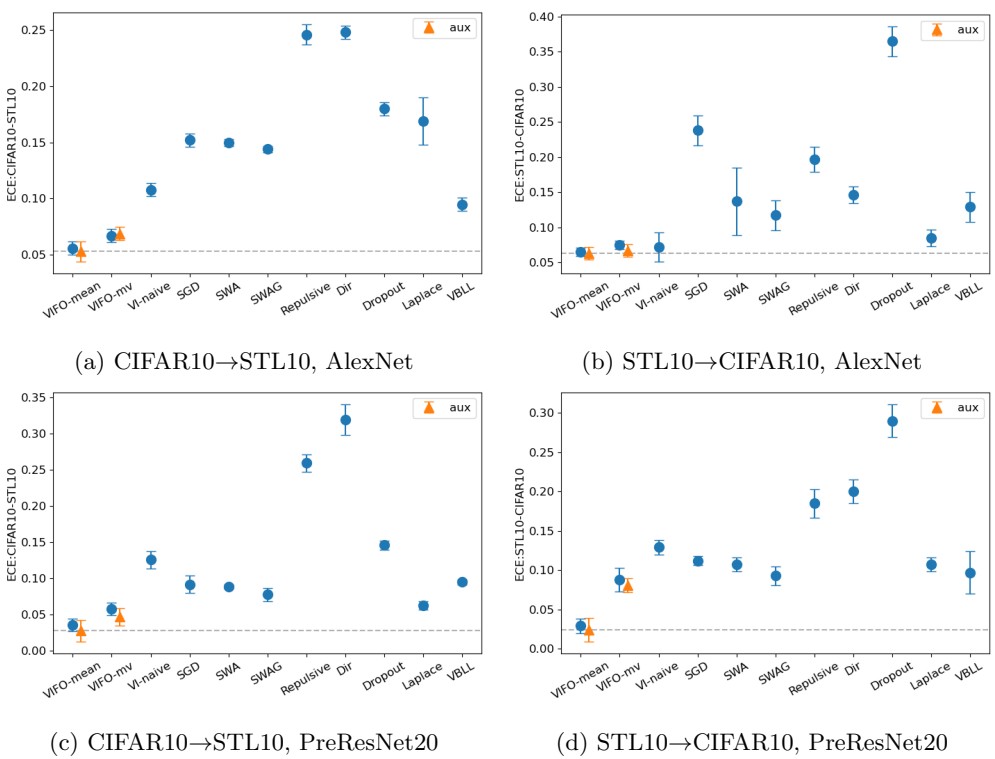

Figure 4: ECE (↓) on AlexNet and PreResNet20 under data shift. Dashed line indicates the best performance of VIFO. Numerical results are listed in the Appendix.

in Fig. 3 and Fig. E.2, repulsive ensembles and the Dirichlet method have poor performance in terms of log loss due to underconfident predictions. Hence they achieve high entropy by sacrificing in distribution performance whereas VIFO performs well. Further, we observe from Fig. 5 that auxiliary training greatly improve the performance of VIFO on PreResNet20. Auxiliary training only has a small impact on VIFO with AlexNet (see Fig. E.4) but VIFO already performs well without auxiliary training in this case.

**AUROC** We use maximum probability of the categorical predictive distribution as the criterion to separate in-distribution and OOD data and compute the area under the ROC curve (Malinin & Gales, 2018). AUROC overcomes the drawbacks of ECE and entropy because a trivial model cannot yield the best performance. Detailed comparison plots are in given in Fig. E.5 and Fig. E.6 in the Appendix. We first note that, as above, auxiliary training improves the performance on PreResNet20 but not significantly on AlexNet. We found that there is no single method that consistently outperforms all other methods. Instead, for better visualization, we show the comparison of VIFO-mean and VIFO-mv with other methods in Fig. 6. For each baseline, we count the number of experiments that VIFO performs better and get the corresponding proportion. We observe that overall, VIFO-mv is better than all other methods except the Dirichlet method and that it ranks better than VIFO-mean. Though the Dirichlet method performs better than VIFO on OOD data, its poor in-distribution performance makes it less desirable. On the other hand, VIFO outperforms all other baselines for OOD and has strong in distribution performance and hence give better overall predictions.

## 7 Conclusion

In Bayesian neural networks, the distribution of the last layer directly affects the predictive distribution. Motivated by this fact, we proposed variational inference on the final-layer output, VIFO, that uses a neural network to directly learn the mean and variance of the last layer. We showed that VIFO can match the expressive power of VI in linear cases with a strong prior but that in general it provides a less expressive model.

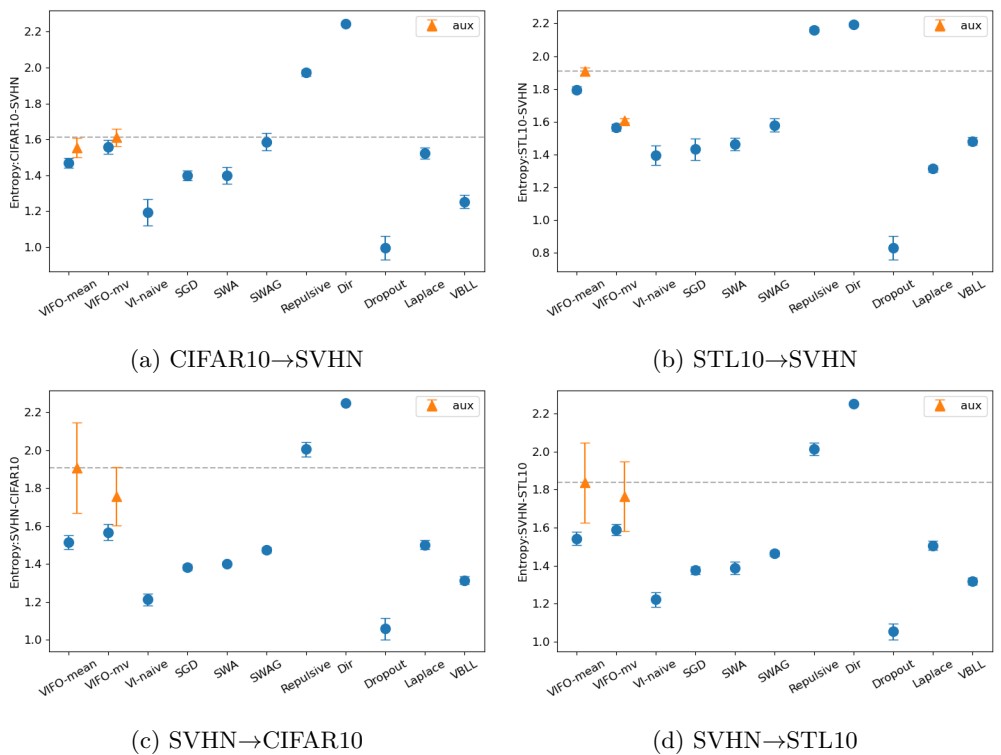

(a) CIFAR10→SVHN

(b) STL10→SVHN

(c) SVHN→CIFAR10

(d) SVHN→STL10

Figure 5: Entropy (↑) on PreResNet20.

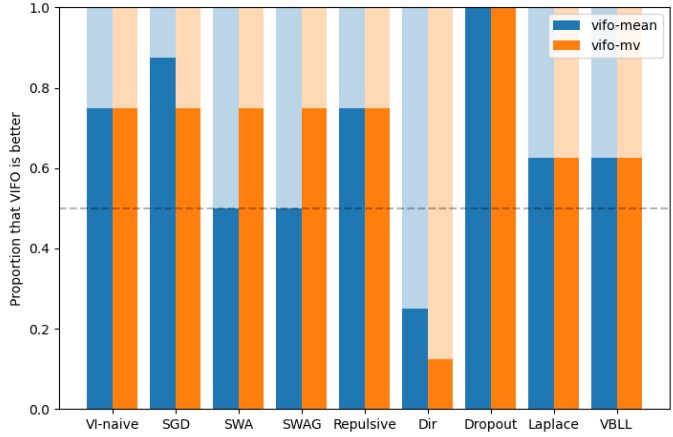

Figure 6: Comparison of VIFO with all other methods in terms of AUROC on OOD data. Y-axis is the proportion of experiments that VIFO is better than other methods. Exact AUROC values are provided in the appendix.

On the other hand the simplicity of the model enables fast training of ensembles of VIFO and facilitates convergence analysis through Rademacher bounds. In addition, VIFO can be derived as a non-standard variational lower bound, which provides an approximation for the last layer. This connection allowed us to derive better regularizations for VIFO by using collapsed variational inference over a hierarchical prior. Since VIFO treats each input separately, we can incorporate auxiliary inputs to help the model distinguish in-distribution and out-of-distribution data. Empirical evaluation highlighted that ensembles of VIFO are competitive with or outperform other methods in terms of in-distribution loss and out-of-distribution data

detection. Hence VIFO gives a new attractive approach for approximate inference in Bayesian models. The efficiency of VIFO also means faster test time predictions which can be important when deploying Bayesian models for real-time applications. Future work could explore more informative auxiliary input to improve the performance of VIFO, and investigate the connections to variational inference in functional space that induces more complex priors.

## Acknowledgments

This work was partly supported by NSF under grants 1906694 and 2246261. Some of the experiments in this paper were run on the Big Red computing system at Indiana University, supported in part by Lilly Endowment, Inc., through its support for the Indiana University Pervasive Technology Institute.

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

# A  Proofs

## A.1  Proofs in Section 3

*Proof of Theorem 3.1.* Assume $N > d$. Note that with the the correlated prior and posterior the covariance function is rank deficient so we have to interpret inverses and determinants appropriately. Here we use

pseudo inverse and pseudo determinant. The VIFO objective is:

$$\sum_{i=1}^{N} \left\{ \mathbb{E}_{q(z|x_i)}[\log p(y_i|z)] \right\} - \text{KL}(q(z|X_N)||p(z|X_N)) \tag{11}$$

$$= \sum_{i=1}^{N} \left\{ \mathbb{E}_{q(z|x_i)}[\log p(y_i|z)] \right\} - \frac{1}{2}\text{tr}((X_N^\top S_0 X_N)^{-1}(X_N^\top V X_N)) + \frac{N}{2}$$

$$+ \frac{1}{2} \log |(X_N^\top S_0 X_N)^{-1}(X_N^\top V X_N)| - \frac{1}{2}(w^\top X_N - m_0^\top X_N)(X_N^\top S_0 X_N)^{-1}(w^\top X_N - m_0^\top X_N)^\top. \tag{12}$$

First consider the loss term. Let $L$ be the Cholesky decomposition of $V$, i.e. $V = LL^\top$. By reparametrization, for $\epsilon \sim \mathcal{N}(0, I_d)$, $w^\top x_i + x_i^\top L\epsilon \sim \mathcal{N}(w^\top x_i, x_i^\top LL^\top x_i)$ and thus

$$\mathbb{E}_{q(z|x_i)}[\log p(y_i|z)] = \mathbb{E}_{\epsilon \sim \mathcal{N}(0, I_d)}[\log p(y_i|w^\top x_i + x_i^\top L\epsilon)]$$

$$= \mathbb{E}_{\epsilon \sim \mathcal{N}(0, I_d)}[\log p(y_i|(w + L\epsilon)^\top x_i)]$$

$$= \mathbb{E}_{\theta \sim \mathcal{N}(w, LL^\top)}[\log p(y_i|\theta^\top x_i)], \tag{13}$$

where the last equality uses reparametrization in a reverse order. By aligning $w = m$ and $V = LL^\top = S$, we recognize that Eq equation 13 is exactly the loss term in Eq equation 9. Thus the low-dimensional posterior on $z$ yields the same loss term as the high-dimensional posterior over $W$.

For the regularization, we use the pseudo inverse derivation from Eq (224) of Petersen & Pedersen (2012), where for $A = CD$ we have $A^+ = D^\top (DD^\top)^{-1}(C^\top C)^{-1}C^\top$ to get

$$(X_N^\top S_0 X_N)^{-1} = X_N^\top (X_N X_N^\top)^{-1} S_0^{-1}(X_N X_N^\top)^{-1} X_N$$

and the same for $V$. Thus,

$$(X_N^\top S_0 X_N)^{-1}(X_N^\top V X_N) = X_N^\top (X_N X_N^\top)^{-1} S_0^{-1}(X_N X_N^\top)^{-1} X_N X_N^\top V X_N$$

$$= X_N^\top (X_N X_N^\top)^{-1} S_0^{-1} V X_N,$$

$$\text{tr}|X_N^\top (X_N X_N^\top)^{-1} S_0^{-1} V X_N| = \text{tr}|X_N X_N^\top (X_N X_N^\top)^{-1} S_0^{-1} V|$$

$$= \text{tr}(S_0^{-1} V),$$

and

$$(w^\top X_N - m_0^\top X_N)(X_N^\top S_0 X_N)^{-1}(w^\top X_N - m_0^\top X_N)^\top$$

$$= (w - m_0)^\top X_N (X_N^\top (X_N X_N^\top)^{-1} S_0^{-1}(X_N X_N^\top)^{-1} X_N) X_N^\top (w - m_0)$$

$$= (w - m_0)^\top (X_N X_N^\top)(X_N X_N^\top)^{-1} S_0^{-1}(X_N X_N^\top)^{-1}(X_N X_N^\top)(w - m_0)$$

$$= (w - m_0)^\top S_0^{-1}(w - m_0).$$

For the The log-determinant term we use the pseudo-determinant (Minka, 2001), which is the product of non-zero eigenvalues. Let $(\lambda_i, u_i)_{i=1}^d$ be the set of eigenvalues and eigenvectors of $S_0^{-1}V$, i.e., $S_0^{-1}V u_i = \lambda u_i$, and let $X_N^\ddagger = X_N^\top (X_N X_N^\top)^{-1}$ denote the pseudo inverse of $X_N$, then

$$(X_N^\ddagger S_0^{-1} V X_N) X_N^\ddagger u_i = X_N^\ddagger S_0^{-1} V u_i = \lambda X_N^\ddagger u_i, \tag{14}$$

thus $(\lambda_i, X_N^\ddagger u_i)_{i=1}^d$ is the eigenvalues and eigenvectors of $X_N^\top (X_N X_N^\top)^{-1} S_0^{-1} V X_N$. Since the rank of this matrix is at most $d$, other eigenvalues are 0 and the pseudo determinant is $\prod_{i=1}^d \lambda_i$, which is exactly the determinant of $S_0^{-1}V$. Then the regularization term in equation 9 can be simplified to:

$$-\text{KL}(q(z|X_N)||p(z|X_N)) = -\frac{1}{2}\text{tr}(S_0^{-1}V) + \frac{1}{2}\log |S_0^{-1}V| - \frac{1}{2}(w - m_0)^\top S_0^{-1}(w - m_0) + \frac{N}{2}. \tag{15}$$

By aligning $w = m, V = S$, we can seet that equation 15 is exactly the regularizer in equation 9 ignoring the constant. $\qquad\square$

**Note for the case** $K > 1$**:** Let $\theta \in \mathbb{R}^{d \times K}$. For VI, we make a mean field assumption with $q(\theta_k) = \mathcal{N}(\theta_k | m_k, S_k)$ and $q(\theta) = \prod_{k=1}^{K} q(\theta_k)$, where $\theta_k$ is the $k$-th column of $\theta$. For VIFO, using mean field let $q(z_k | x) = \mathcal{N}(z_k | w_k^\top x, x^\top V_k x)$ and $q(z | x) = \prod_{k=1}^{K} q(z_k | x)$. By aligning $w_k = m_k$ and $V = S_k$, we can find $\mathbb{E}_{q(z|x_i)}[\log p(y_i | z_1, \ldots z_K)] = \mathbb{E}_{q(\theta)}[\log p(y_i | (\theta_1^\top x_i, \ldots, \theta_K^\top x_i))]$, and

$$\mathrm{KL}(q(\theta)||p(\theta)) = \sum_k \mathrm{KL}(q(\theta_k), p(\theta_k)) \doteq \sum_k \mathrm{KL}(q(z_k | X_N) || p(z_k | X_N)), \tag{16}$$

where the second $\doteq$ means equivalence ignoring a constant difference.

*Proof of Theorem 3.2.* Consider a neural network with one single hidden layer, denote the weights of the first layer as $u$, and the weights of the second layer as $w$. Thus, the $k$-th output can be computed as:

$$z^{(k)} = \sum_{i=1}^{I} w_{k,i} \psi \left( \sum_{d=1}^{D} u_{i,d} x_d \right),$$

where $I$ is the size of the hidden layer, $D$ is the input size and $\psi(a) = \max(0, a)$ is the ReLU activation function. We further simplify the setting by considering the special case where only $x_1$ is non-zero and $I = 1$. Then the $k$-th output becomes:

$$z^{(k)} = w_{k,1} \psi(u_{1,1} x_1).$$

Consider a distribution $q(w_{k,i}) = \mathcal{N}(\bar{w}_{k,i}, \sigma_w^2), q(u_{i,d}) = \mathcal{N}(\bar{u}_{i,d}, \sigma_u^2)$. Then if $x_1 \geq 0$,

$$\mathbb{E}_{q(w)q(u)} \left[ z^{(k)} \right] = \mathbb{E}_{w,u} \left[ w_{k,1} \psi(u_{1,1} x_1) \right]$$

$$= \bar{w}_{k,1} \left( \bar{u}_{1,1} + \frac{\phi\left(-\frac{\bar{u}_{1,1}}{\sigma_u}\right)}{1 - \Phi\left(-\frac{\bar{u}_{1,1}}{\sigma_u}\right)} \sigma_u \right) \left( 1 - \Phi\left(-\frac{\bar{u}_{1,1}}{\sigma_u}\right) \right) x_1; \tag{17}$$

if $x_1 < 0$, then

$$\mathbb{E}_{q(w)q(u)} \left[ z^{(k)} \right] = \mathbb{E}_{w,u} \left[ w_{k,1} \psi(u_{1,1} x_1) \right]$$

$$= \bar{w}_{k,1} \left( \bar{u}_{1,1} - \frac{\phi\left(-\frac{\bar{u}_{1,1}}{\sigma_u}\right)}{\Phi\left(-\frac{\bar{u}_{1,1}}{\sigma_u}\right)} \sigma_u \right) \Phi\left(-\frac{\bar{u}_{1,1}}{\sigma_u}\right) x_1, \tag{18}$$

where $\phi$ and $\Phi$ are the pdf and cdf of standard normal distribution and we directly use the expectation of the truncated normal distribution. Now consider $\tilde{w}$ and $\tilde{u}$ that aim to recover (17) and (18). If $\tilde{u}_{1,1} \geq 0$, it cannot successfully recover (18) because the ReLU activation will have 0 when $x_1 < 0$ so that it cannot recover (18); if $\tilde{u}_{1,1} < 0$, for the same reason it cannot recover (17). $\qquad\square$

## A.2   Proofs in Section 4

**Verifying Assumption 4.1:** We next verify that Assumption 4.1 holds for classification and (with a modified loss) for regression.

For $K$-classification, $z$ is $K$-dimensional and the negative log-likelihood is

$$-\log p(y = k | z) = -\log \frac{\exp(z_k)}{\sum_{i=1}^{K} \exp(z_i)} = -z_k + \log \sum_{i=1}^{K} \exp(z_i)$$

which is 1-Lipschitz in $z$.

For regression, $z = (m, l)$ is 2-dimensional, and the negative log-likelihood is:

$$-\log p(y | z) = \frac{1}{2}(y - m)^2 \exp(-l) + \frac{1}{2} l.$$

Neither the quadratic function nor exponential function is Lipschitz. But we can replace the unbounded quadratic function $(y-m)^2$ with a bounded version $\min\{(y-m)^2, B_m^2\}$, and replace the exponential function $\exp(-l)$ with $\min\{\exp(-l), B_l\}$, where $B > 0$, to guarantee the Lipschitzness. Now the negative log-likelihood is:

$$-\log p(y|z) = \frac{1}{2}\min\{(y-m)^2, B_m^2\}\min\{\exp(-l), B_l\} + \frac{1}{2}l,$$

is $(B_m B_l)$-Lipschitz in $m$, $\left(\frac{1}{2} + \frac{1}{2}B_m^2 B_l\right)$-Lipschitz in $l$.

**Verifying Assumption 4.2:** For Assumption 4.2, we can use $g(l) = \log(1 + \exp(l))$ which is 1-Lipschitz. If $g(l) = \exp(l)$ is the exponential function, we can use a bounded variant that satisfies the requirement $g(l) = \max\{\exp(x), B_g\}$.

*Proof of Lemma 4.3.* We prove the lemma for $L = 1$. If this is not the case, we can define $\phi' = \frac{1}{L}\phi$, and use the fact that $R(\phi(A \times B)) \leq LR(\phi'(A \times B))$. Let $C_i = \{(a_1+b_1, \ldots, a_{i-1}+b_{i-1}, \phi'(a_i, b_i), a_{i+1}+b_{i+1}, \ldots, a_N+b_N) : a \in A, b \in B\}$. It suffices to prove that for any set $A, B$ and all $i$ we have $R(C_i) \leq R(A) + R(B)$. Without loss of generality we prove the case for $i = 1$.

$$NR(C_1) = \mathbb{E}_\sigma \left[\sup_{c \in C_1} \sigma_1\phi(a_1, b_1) + \sum_{i=2}^N \sigma_i(a_i + b_i)\right]$$

$$= \frac{1}{2}\mathbb{E}_{\sigma_2,\ldots,\sigma_N}\left[\sup_{a \in A, b \in B}\left(\phi(a_1, b_1) + \sum_{i=2}^N \sigma_i(a_i + b_i)\right)\right.$$

$$\left. + \sup_{a' \in A, b' \in B}\left(-\phi(a_1', b_1') + \sum_{i=2}^N \sigma_i(a_i' + b_i')\right)\right]$$

$$= \frac{1}{2}\mathbb{E}_{\sigma_2,\ldots,\sigma_N}\left[\sup_{a,a' \in A, b,b' \in B}\left(\phi(a_1, b_1) - \phi(a_1', b_1') + \sum_{i=2}^N \sigma_i(a_i + b_i) + \sum_{i=2}^N \sigma_i(a_i' + b_i')\right)\right]$$

$$\leq \frac{1}{2}\mathbb{E}_{\sigma_2\ldots\sigma_N}\left[\sup_{a,a' \in A, b,b' \in B}\left(|a_1 - a_1'| + |b_1 - b_1'| + \sum_{i=2}^N \sigma_i(a_i + b_i) + \sum_{i=2}^N \sigma_i(a_i' + b_i')\right)\right]$$

$$= \frac{1}{2}\mathbb{E}_{\sigma_2,\ldots,\sigma_N}\left[\sup_{a,a' \in A}\left(a_1 - a_1' + \sum_{i=2}^N \sigma_i a_i + \sum_{i=2}^N \sigma_i a_i'\right)\right]$$

$$+ \frac{1}{2}\mathbb{E}_{\sigma_2,\ldots,\sigma_N}\left[\sup_{b,b' \in B}\left(b_1 - b_1' + \sum_{i=2}^N \sigma_i b_i + \sum_{i=2}^N \sigma_i b_i'\right)\right]$$

$$= NR(A) + NR(B).$$

$\square$

*Proof of Theorem 4.5.* We show that the loss is Lipschitz in $f_{W_1}(x)$ and $f_{W_2}(x)$. Fix any $x$, and $W, W'$. We denote the mean and standard deviation of $q_W(z|x)$ by $\mu$ and $s$ and the same for $q_{W'}(z|x)$. We use $\cdot$ for Hadamard product.

$$\mathbb{E}_{q_W(z|x)}[-\log p(y|z)] - \mathbb{E}_{q_{W'}(z|x)}[-\log p(y|z)]$$

$$= \mathbb{E}_{\epsilon \sim \mathcal{N}(0,I)}[\log p(y|\mu' + \epsilon \cdot s') - \log p(y|\mu + \epsilon \cdot s)]$$

$$\leq \mathbb{E}_{\epsilon \sim \mathcal{N}(0,I)}\left[L_0\|(\mu - \mu') + \epsilon \cdot (s - s')\|_2\right] \qquad \text{(Lipschitz)}$$

$$\leq L_0\|\mu - \mu'\|_2 + L_0\mathbb{E}_\epsilon\left[\sqrt{\|\epsilon \cdot (s - s')\|_2^2}\right]$$

$$\leq L_0\|\mu - \mu'\|_2 + L_0\sqrt{\mathbb{E}_\epsilon[\|\epsilon \cdot (s - s')\|_2^2]} \qquad \text{(Jensen's Ineq)}$$

$$= L_0(\|\mu - \mu'\|_2 + \|s - s'\|_2)$$

$$\leq L_0(\|\mu - \mu'\|_1 + \|s - s'\|_1).$$

For the 6th line note that $\mathbb{E}_\epsilon[\|\epsilon \cdot (s-s')\|_2^2] = \mathbb{E}_\epsilon[\sum_i \epsilon_i^2(s_i-s_i')^2] = \sum_i \mathbb{E}_{\epsilon_i \sim \mathcal{N}(0,1)}[\epsilon_i^2(s_i-s_i)^2] = \sum_i (s_i-s_i')^2 = \|s-s'\|_2^2$. The loss function is Lipschitz in $\mu$, which is exactly $f_{W_1}(x)$. Further, $s$ is $L_1$-Lipschitz in the logit $f_{W_2}(x)$, thus, the loss function is $(L_0 \max\{1, L_1\})$-Lipschitz in the concatenation of $f_{W_1}(x)$ and $f_{W_2}(x)$, each of which is of dimension $K$. The theorem now follows from Corollary 4.4. □

# B Derivations of Collapsed Variational Inference

As is shown by Tomczak et al. (2021), for priors and approximate posteriors from the exponential family, we can derive the closed-form solution for the optimal $q^*(\mu_p, \sigma_p^2)$,

$$\log q^*(\mu_p, \sigma_p^2|x) \propto \log p(\mu_p, \sigma_p^2) + \mathbb{E}_{q(z|x)}[\log p(z|\mu_p, \sigma_p^2)], \tag{19}$$

for optimizing $q(\mu_p, \sigma_p^2)$ for every single data. Our derivations follow the methodology of Tomczak et al. (2021) but they are applied on the output $z$ instead of the weights $W$.

## B.1 Learn mean, fix variance

Let $p(z|\mu_p) = \mathcal{N}(z|\mu_p, \gamma I)$, $p(\mu_p) = \mathcal{N}(\mu_p|0, \alpha I)$. Recall that $q(z|x) = \mathcal{N}(\mu_q(x), \text{diag}(\sigma_q^2(x)))$. Then

$$\log q^*(\mu_p|x) \propto \log p(\mu_p) + \mathbb{E}_{q(z|x)}[\log p(z|\mu_p)]$$

$$\propto -\frac{1}{2\alpha}\mu_p^\top \mu_p - \frac{1}{2\gamma}[(\mu_p - \mu_q(x))^\top(\mu_p - \mu_q(x)) + 1^\top \sigma_q^2(x)]$$

$$\propto -\frac{\alpha+\gamma}{2\alpha\gamma}\left(\mu_p - \frac{\alpha}{\alpha+\gamma}\mu_q(x)\right)^\top \left(\mu_p - \frac{\alpha}{\alpha+\gamma}\mu_q(x)\right).$$

Then $q^*(\mu_p) = \mathcal{N}(\frac{\alpha}{\alpha+\gamma}\mu_q(x), \frac{\alpha\gamma}{\alpha+\gamma}I)$. Pluging $q^*$ into the regularizer, the new regularizer becomes

$$\frac{1}{2\gamma}\left[1^\top \sigma_q^2(x) + \frac{\gamma}{\gamma+\alpha}\mu_q(x)^\top \mu_q(x)\right] - \frac{1}{2}1^\top \log \sigma_q^2(x) + \frac{K}{2}\log(\gamma+\alpha) - \frac{K}{2}.$$

## B.2 Learn both mean and variance

Let $p(z|\mu_p, \sigma_p^2) = \mathcal{N}(z|\mu_p, \sigma_p^2)$, $p(\mu_p|\sigma_p^2) = \mathcal{N}(\mu_p|0, \frac{1}{t}\sigma_p^2)$, $p(\sigma_p^2) = \mathcal{IG}(\sigma_p^2|\alpha, \beta)$, where $\mathcal{IG}$ indicates the inverse Gamma distribution. Let $q(\mu_p)$ be a diagonal Gaussian and $q(\sigma_p^2)$ be inverse Gamma. Use $\mu_{p,i}$ and $\sigma_{p,i}$ to denote the $i$-th entry of $\mu_p$ and $\sigma_p$ respectively, then

$$\log q^*(\mu_{p,i}, \sigma_{p,i}^2)$$

$$\propto \log p(\mu_{p,i}, \sigma_{p,i}^2) + \mathbb{E}_{q(z|x)}[\log p(z|\mu_p, \sigma_p^2)]$$

$$\propto -(\alpha + \frac{3}{2})\log \sigma_{p,i}^2 - \frac{2\beta + t\mu_{p,i}^2}{2\sigma_{p,i}^2} - \frac{1}{2}\log \sigma_{p,i}^2 - \frac{1}{2}\frac{(\mu_{p,i} - \mu_{q,i}(x))^2}{\sigma_{p,i}^2} - \frac{1}{2}\frac{\sigma_{q,i}^2(x)}{\sigma_{p,i}^2}$$

$$\propto -(\alpha + 2)\log \sigma_{p,i}^2 - \frac{1}{2\sigma_{p,i}^2}\left(2\left(\beta + \frac{t}{2(t+1)}\mu_{q,i}(x)^2 + \frac{1}{2}\sigma_{q,i}^2(x)\right) + (t+1)\left(\mu_{p,i} - \frac{\mu_{q,i}(x)}{t+1}\right)^2\right)$$

follows the normal-inverse-gamma distribution. Thus $q^*(\mu_p|x) = \mathcal{N}(\mu_p|\frac{1}{t+1}\mu_q(x), \frac{1}{t+1}\sigma_p^2)$ and $q^*(\sigma_p^2|x) = \mathcal{IG}(\sigma_p^2|(\alpha + \frac{1}{2})1, \beta + \frac{t}{2(t+1)}\mu_q(x)^2 + \frac{1}{2}\sigma_q^2(x))$. Then the regularizer becomes

$$(\alpha + \frac{1}{2})1^\top \log\left[\beta 1 + \frac{t}{2(1+t)}\mu_q(x)^2 + \frac{1}{2}\sigma_q^2(x)\right] - \frac{1}{2}1^\top \log \sigma_q^2(x). \tag{20}$$

### B.3 Empirical Bayes

Let $p(\sigma_p^2) = \mathcal{IG}(\sigma_p^2|\alpha, \beta)$, $p(z|\sigma_p^2) = \mathcal{N}(z|0, \sigma_p^2 I)$, and let $q(\sigma_p^2)$ be a delta distribution. Then

$$
\mathrm{KL}(q(z|x)||p(z|\sigma_p^2)) - \log p(\sigma_p^2)
$$
$$
= \frac{1}{2}\left[ K \log \sigma_p^2 - 1^\top \log \sigma_q^2(x) - K + \frac{1^\top \sigma_q^2(x)}{\sigma_p^2} + \frac{\mu_q(x)^\top \mu_q(x)}{\sigma_p^2} \right] + (\alpha + 1)\log \sigma_p^2 + \frac{\beta}{\sigma_p^2}.
$$

By taking the derivatives of the above equation with respect to $\sigma_p^2$ and solving, we obtain the optimal $\sigma_p^2 = \frac{\mu_q(x)^\top \mu_q(x) + 1^\top \sigma_q^2(x) + 2\beta}{K + 2\alpha + 2}$. If we plug this back into the KL term, we get the regularizer:

$$
\frac{1}{2}\left[ K \log \frac{\mu_q(x)^\top \mu_q(x) + 1^\top \sigma_q^2(x) + 2\beta}{K + 2\alpha + 2} - 1^\top \log |\sigma_q^2(x)| \right]
$$
$$
- \frac{K}{2} + \frac{1}{2}\frac{(K + 2\alpha + 2)(\mu_q(x)^\top \mu_q(x) + 1^\top \sigma_q^2(x))}{\mu_q(x)^\top \mu_q(x) + 1^\top \sigma_q^2(x) + 2\beta}. \tag{21}
$$

However, if we include the negative log-prior term $(\alpha + 1)\log \frac{\mu_q(x)^\top \mu_q(x) + 1^\top \sigma_q^2(x) + 2\beta}{K + 2\alpha + 2} + \beta \frac{K + 2\alpha + 2}{\mu_q(x)^\top \mu_q(x) + 1^\top \sigma_q^2(x) + 2\beta}$, adding them up we will have

$$
\frac{1}{2}(K + 2\alpha + 2)\log \frac{\mu_q(x)^\top \mu_q(x) + 1^\top \sigma_q^2(x) + 2\beta}{K + 2\alpha + 2} - 1^\top \log \sigma_q^2(x) + \mathrm{const},
$$

which highly reduces the complexity of the regularizer. This performs less well in practice and therefore we follow Wu et al. (2019) and use equation 21.

## C  Optimizing the Variational Distribution for All Data

In the previous section we show the derivation of collapsed variational inference where $q^*(\mu_p, \sigma_p^2)$ is optimized for every data point $x$. In this section we show how to optimize $q(\mu_p, \sigma_p^2)$ for all data and obtain different regularizers to the ones mentioned in the above section. These perform less well in practice but we include them here for completeness. The closed-form solution for $q^*(\mu_p, \sigma_p^2)$ for all data is

$$
\log q^*(\mu_p, \sigma_p^2) \propto \frac{1}{N}\sum_{(x,y)\in\mathcal{D}}\left\{ \log p(\mu_p, \sigma_p^2) + \mathbb{E}_{q(z|x)}[\log p(z|\mu_p, \sigma_p^2)] \right\}. \tag{22}
$$

### C.1  Learn mean, fix variance, optimize for all data

Let $p(z|\mu_p) = \mathcal{N}(z|\mu_p, \gamma)$, $p(\mu_p) = \mathcal{N}(\mu_p|0, \alpha)$. Given a dataset $\mathcal{D} = \{(x, y)\}$, we can get one optimal $q^*(\mu_p)$ for all data. According to equation 22,

$$
\log q^*(\mu_p) \propto \frac{1}{N}\sum_{(x,y)\in\mathcal{D}}\left\{ \log p(\mu_p) + \mathbb{E}_{q(z|x)}[\log p(z|\mu_p)] \right\}
$$
$$
\propto -\frac{1}{2\alpha}\mu_p^\top \mu_p - \frac{1}{2\gamma N}\sum_{(x,y)\in\mathcal{D}}((\mu_p - \mu_q(x))^\top(\mu_p - \mu_q(x)) + 1^\top \sigma_q^2(x))
$$
$$
\propto -\frac{\alpha + \gamma}{2\alpha\gamma}\left(\mu_p - \frac{1}{N}\sum_{(x,y)\in\mathcal{D}}\mu_q(x)\right)^\top\left(\mu_p - \frac{1}{N}\sum_{(x,y)\in\mathcal{D}}\mu_q(x)\right).
$$

Then the optimal $q^*(\mu_p) = \mathcal{N}(\frac{\alpha}{\alpha+\gamma} \frac{1}{N} \sum_x \mu_q(x), \frac{\alpha\gamma}{\alpha+\gamma} I)$. Let $\bar{\mu}_q = \frac{1}{N} \sum_x \mu_q(x)$. The regularizer now is:

$$\sum_{(x,y)} \left\{ \mathbb{E}_{q(\mu_p)}[\text{KL}(q(z|x)||p(z|\mu_p, \gamma))] + \text{KL}(q(\mu_p)||p(\mu_p)) \right\}$$

$$= \sum_{(x,y)} \left\{ \mathbb{E}_{q(\mu_p)} \left[ K \log \gamma - 1^\top \log \sigma_q^2(x) - K + \frac{1}{\gamma} 1^\top \sigma_q^2(x) - \frac{1}{\gamma} (\mu_q(x) - \mu_p)^\top (\mu_q(x) - \mu_p) \right] \right\}$$

$$+ \frac{N}{2} \left[ K \log \frac{\alpha+\gamma}{\gamma} - K + K \frac{\gamma}{\gamma+\alpha} + \frac{\alpha^2}{(\alpha+\gamma)^2} \bar{\mu}_q^\top \bar{\mu}_q \right]$$

$$= \sum_{(x,y)} \left\{ \frac{1}{2\gamma} (1^\top \sigma_q^2(x) + \mu_q(x)^\top \mu_q(x)) - \frac{1}{2} 1^\top \log \sigma_q^2(x) \right\} - \frac{N}{2} \left( \frac{1}{\gamma} - \frac{1}{\alpha+\gamma} \right) \bar{\mu}_q^\top \bar{\mu}_q + \frac{NK}{2} \log(\alpha+\gamma) - \frac{NK}{2}. \tag{23}$$

We refer to this method as "VIFO-mean_all".

## C.2 Learn both mean and variance, optimize mean for single data, and variance for all data

Let $p(z|\mu_p, \sigma_p^2) = \mathcal{N}(z|\mu_p, \sigma_p^2)$, $p(\mu_p|\sigma_p^2) = \mathcal{N}(\mu_p|0, \frac{1}{t}\sigma_p^2)$, $p(\frac{1}{\sigma_p^2}) = \mathcal{IG}(\frac{1}{\sigma_p^2}|\alpha, \beta)$. Consider that

$$\log p(\mu_{p,i}, \sigma_{p,i}^2) + \mathbb{E}_{q(z|x)}[\log p(z|\mu_{p,i}, \sigma_{p,i}^2)] \tag{24}$$

$$= \log p(\mu_{p,i}|\sigma_{p,i}^2) + \log p(\sigma_{p,i}^2) + \mathbb{E}_{q(z|x)}[\log p(z|\mu_{p,i}, \sigma_{p,i}^2)] \tag{25}$$

$$\propto -\frac{t}{2} \frac{\mu_{p,i}^2}{\sigma_{p,i}^2} - \frac{1}{2} \log \sigma_{p,i}^2 - (\alpha+1) \log \sigma_{p,i}^2 - \frac{\beta}{\sigma_{p,i}^2} - \frac{1}{2} \log \sigma_{p,i}^2 - \frac{1}{2\sigma_{p,i}^2} ((\mu_{p,i} - \mu_{q,i}(x))^2 + \sigma_{q,i}^2(x)) \tag{26}$$

$$= -\frac{t}{2} \frac{\mu_{p,i}^2}{\sigma_{p,i}^2} - \log \sigma_{p,i}^2 - (\alpha+1) \log \sigma_{p,i}^2 - \frac{\beta}{\sigma_{p,i}^2} - \frac{1}{2\sigma_{p,i}^2} ((\mu_{p,i} - \mu_{q,i}(x))^2 + \sigma_{q,i}^2(x)), \tag{27}$$

$$= -\frac{t+1}{2\sigma_{p,i}^2} \left( \mu_{p,i} - \frac{1}{t+1} \mu_{q,i}(x) \right)^2 - \frac{t\mu_{q,i}^2(x)}{2(t+1)\sigma_{p,i}^2} - (\alpha+2) \log \sigma_{p,i}^2 - \frac{\beta}{\sigma_{p,i}^2} - \frac{\sigma_{q,i}^2(x)}{2\sigma_{p,i}^2} \tag{28}$$

Then by extracting the $\mu_p$ part from equation 28, we have

$$\log q^*(\mu_{p,i}|\sigma_{p,i}^2, x) \propto -\frac{t+1}{2\sigma_{p,i}^2} \left( \mu_{p,i} - \frac{1}{t+1} \mu_{q,i}(x) \right)^2,$$

and thus $q^*(\mu_p|x, \sigma_p^2) = \mathcal{N}(\mu_p|\frac{1}{t+1}\mu_q(x), \frac{1}{t+1}\sigma_p^2)$. Then we try to marginalize out $\mu_p$ to compute $q^*(\sigma_p^2)$.

$$\log q^*(\sigma_{p,i}^2) \propto \frac{1}{N} \sum_{(x,y)\in\mathcal{D}} \log \int \exp \left( \log p(\mu_{p,i}, \sigma_{p,i}^2) + \mathbb{E}_{q(z|x)}[\log p(z|\mu_{p,i}, \sigma_{p,i}^2)] \right) d\mu_{p,i}$$

$$\propto \frac{1}{N} \sum_{x,y\in\mathcal{D}} \left\{ \frac{1}{2} \log \int \exp \left( -\frac{t+1}{2\sigma_{p,i}^2} \left( \mu_{p,i} - \frac{1}{t+1} \mu_{q,i}(x) \right)^2 \right) d\mu_{p,i} \right.$$

$$\left. -\frac{t\mu_{q,i}^2(x)}{2(t+1)\sigma_{p,i}^2} - (\alpha+2) \log \sigma_{p,i}^2 - \frac{\beta}{\sigma_{p,i}^2} - \frac{\sigma_{q,i}^2(x)}{2\sigma_{p,i}^2} \right\}$$

$$= -(\alpha+2) \log \sigma_{p,i}^2 - \frac{\beta}{\sigma_{p,i}^2} + \frac{1}{N} \sum_{(x,y)\in\mathcal{D}} \left( \frac{1}{2} \log \frac{2\pi\sigma_{p,i}^2}{t+1} - \frac{\sigma_{q,i}^2(x)}{2\sigma_{p,i}^2} - \frac{t\mu_{q,i}^2(x)}{2(t+1)\sigma_{p,i}^2} \right)$$

$$= -(\alpha+\frac{3}{2}) \log \sigma_{p,i}^2 - \frac{\beta + \frac{t}{2(t+1)} \frac{1}{N} \sum \mu_{q,i}^2(x) + \frac{1}{2N} \sum \sigma_{q,i}^2(x)}{\sigma_{p,i}^2},$$

and $q^*(\sigma_p^2) = \mathcal{IG}(\sigma_p^2|(\alpha + \frac{1}{2})1, \beta + \frac{t}{2(t+1)} \frac{1}{N} \sum_x \mu_q(x)^2 + \frac{1}{2} \frac{1}{N} \sum_x \sigma_q^2(x))$. Let $\tilde{\mu}_q = \sqrt{\frac{1}{N} \sum_x \mu_q(x)^2}$ and $\tilde{\sigma}_q = \sqrt{\frac{1}{N} \sum_x \sigma_q(x)^2}$, then the regularizer becomes:

$$(\alpha + \frac{1}{2})N1^\top \log \left[\beta 1 + \frac{t}{2(1+t)} \tilde{\mu}_q^2 + \frac{1}{2} \tilde{\sigma}_q^2\right] - \sum_{(x,y)} \frac{1}{2} 1^\top \log \sigma_q^2(x) \tag{29}$$

$$+ KN \log \frac{\Gamma(\alpha)}{\Gamma(\alpha + \frac{1}{2})} - NK\alpha \log \beta + \frac{NK}{2} \log \frac{t+1}{t} - \frac{NK}{2}. \tag{30}$$

We refer to this method as "VIFO-mv_all".

### C.3 Empirical Bayes for all data

If we optimize $\sigma_p^2$ for all data, then we have

$$\sum_{(x,y)\in\mathcal{D}} \left\{ \text{KL}(q(z|x)||p(z|\sigma_p^2)) - \log p(\sigma_p^2) \right\}$$

$$= \sum_{(x,y)\in\mathcal{D}} \left\{ \frac{1}{2} \left[ K \log \sigma_p^2 - 1^\top \log \sigma_q^2(x) - K + \frac{1^\top \sigma_q^2(x)}{\sigma_p^2} + \frac{\mu_q(x)^\top \mu_q(x)}{\sigma_p^2} \right] + (\alpha + 1) \log \sigma_p^2 + \frac{\beta}{\sigma_p^2} \right\}$$

and the optimal variance being $\frac{\tilde{\mu}_q^\top \tilde{\mu}_q + 1^\top \tilde{\sigma}_q^2 + 2\beta}{K + 2\alpha + 2}$ where $\tilde{\mu}_q = \sqrt{\frac{1}{N} \sum_x \mu_q(x)^2}$ and $\tilde{\sigma}_q = \sqrt{\frac{1}{N} \sum_x \sigma_q(x)^2}$. The objective is:

$$\frac{NK}{2} \log \frac{2\beta + \tilde{\mu}_q^2 + \tilde{\sigma}_q^2}{K + 2\alpha + 2} - \frac{1}{2} \sum_x 1^\top \log \sigma_q(x)^2 - \frac{NK}{2}$$

$$+ \frac{1}{2} \frac{K + 2\alpha + 2}{2\beta + \tilde{\mu}_q^2 + \tilde{\sigma}_q^2} \sum_x (\mu_q(x)^\top \mu_q(x) + 1^\top \sigma_q^2(x)).$$

This method is called "VIFO-eb_all".

## D Experimental Details

### D.1 Experiments on Artificial Dataset

To generate Fig. 1 and Fig. 2, we generate 100 training data points $y = 2\sin x + 0.1\epsilon, \epsilon \sim \mathcal{N}(0,1)$, where $x_{\text{train}} \in [-\frac{3}{4}\pi, -\frac{1}{2}\pi] \cup [\frac{1}{2}\pi, \frac{3}{4}\pi]$ and $x_{\text{test}} \in [-\pi, \pi]$. We use a multilayer perceptron neural network with 5 layers, each layer containing 50 hidden units to fit the data. For VI, we pick the prior standard deviation to be 1.0 and for VIFO-mean, we select $\gamma = 0.3, \frac{\gamma}{\alpha+\gamma} = 0.05$. For both models, we select the regularization parameter $\eta = 0.1$ and for VIFO we choose $\eta_{\text{aux}} = 1.0$. For linear regression, we can explicitly compute the predicted variance if we use exponential function as the link function. Suppose $p(y|z) = \mathcal{N}(y \mid m, \exp(l))$ and $p(m \mid x) = \mathcal{N}(m \mid \mu_m, \sigma_m^2), p(l \mid x) = \mathcal{N}(l \mid \mu_l, \sigma_l^2)$, then $p(y \mid x) = \mathcal{N}(y \mid \mu_m, \sigma_m^2 + \exp(m_l + \sigma_l^2/2))$. See Appendix B.3 of Wu et al. (2019) for the derivation. To visualize the predictive distribution inducted by the prior, we draw multiple $z$'s from prior, then draw multiple $y$'s from likelihood $p(y|z)$ and plot them in Fig. 2.

### D.2 Experiments on Large Image Datasets

In this section we elaborate the experimental details, including the choice of hyperparameters, learning rates and the number of training epochs.

**Number of training epochs:** We train all methods in 500 epochs.

**Learning rate:** For all methods other than SGD, SWA and SWAG, we use the Adam optimizer with learning rate 0.001.

**VI and VIFO:** We first list the choices of the variance for naive variational methods. The choice of prior variance significantly affects the performance. For image datasets with complex neural networks, the total prior variance of VI grows with the number of parameters so we have to pick a small variance and we use 0.05 following the setting of Wilson et al. (2022). Since VIFO samples in the output space which is small, using 0.05 regularizes too strongly and we therefore set a larger value of 1 for the variance.

For collapsed variational inference, we pick $\gamma = 0.3$, $\alpha_{reg} = \frac{\gamma}{\alpha+\gamma} = 0.05$ for learn-mean regularizer (VI-mean, VIFO-mean, VIFO-mean-all) and $\alpha = 0.5, \beta = 0.01, \delta = \frac{t}{1+t} = 0.1$ for learn-mean-variance regularizer (VI-mv, VIFO-mv, VIFO-mv_all), which exactly follows Tomczak et al. (2021). We pick $\alpha = 4.4798$ and $\beta = 10$ for empirical Bayes (VI-eb, VIFO-eb, VIFO-eb_all). The choice of $\alpha$ in empirical Bayes follows Wu et al. (2019) but the choice of $\beta$ is unclear in Wu et al. (2019) so we just perform a simple search from $\{1, 10, 100\}$ and set $\beta = 10$ that yields the best result.

For both VI and VIFO, the regularization parameter $\eta$ is fixed at 0.1.

**Hybrid Methods:** The hybrid methods (SGD, SWA and SWAG) are not very stable so we have to tune learning rates carefully for each dataset. We choose the momentum to be 0.9 for all cases and list all other information in Table D.1. Notice that it is hard to train the hybrid methods on SVHN using AlexNet, so we initialize with a pre-trained model that is trained with a larger learning rate 0.1 to find a region with lower training loss, and then continue to optimize with the parameters listed in Table D.1.

**Dropout:** For Dropout we add a Dropout layer following each activation layer in the base model and set the Dropout probability $p = 0.1$.

**Repulsive Ensembles:** Repulsive ensembles run multiple copies of the base model with a kernel base penalty to make sure the models are diverse. We use RBF kernel with lengthscale being the median of the square of the norm.

**Dirichlet:** Dirichlet-based models are deterministic and they interpret the output of the last layer as the parameters of dirichlet distributions, i.e., $\boldsymbol{\alpha}(x) = g(f_W(x))$, where $g$ maps the output to positive real numbers. Hence we run the Dirichlet models with the setting of the base model. We next explain the setting of hyperparameters. As discussed by Bengs et al. (2022), the models of Sensoy et al. (2018); Charpentier et al. (2020) implicitly perform variational inference:

$$\mathbf{p} \sim \text{Dir}(\boldsymbol{\alpha}_0), \quad y|\mathbf{p} \sim \text{Cat}(\mathbf{p}), \tag{31}$$

and the ELBO becomes

$$\log p(y|x) \geq \mathbb{E}_{q(\mathbf{p}|x)}[\log p(y|\mathbf{p})] - \text{KL}(q(\mathbf{p}|x)||\text{Dir}(\mathbf{p}|\boldsymbol{\alpha}_0)), \tag{32}$$

where $q(\mathbf{p}|x) = \text{Dir}(\mathbf{p}|\boldsymbol{\alpha}(x))$. In the experiments, following Sensoy et al. (2018); Bengs et al. (2022), we use a uniform prior with $\boldsymbol{\alpha}_0 = [1, \dots, 1]$. As in VI and VIFO, we pick the regularization parameter for KL divergence to be 0.1.

**Last layer Laplace:** We first train a neural network to obtain a MAP solution with a prior variance of 0.05. Then, we use the code from Daxberger et al. (2021) to optimize the prior precision hyperparameter through post-hoc marginal likelihood maximization.

**VBLL:** We adapt the code from Harrison et al. (2024) and utilize the default hyperparameters. We choose the discriminative classification setting, as it yields the best OOD performance according to Harrison et al. (2024).

Table D.1: The parameters for running the hybrid algorithms. "lr"-learning rate, "wd"-weight decay, "swag_lr"-the learning rate after we start collecting models in SWA and SWAG algorithms, "swag_start"-the epochs when we start to collect models, "epochs"-the number of training epochs.

|  | lr | wd | swag_lr | swag_start | epochs |
|---|---|---|---|---|---|
| CIFAR10 / CIFAR100 | 0.05 | 0.0001 | 0.01 | 161 | 500 |
| SVHN* | 0.001 | 0.0001 | 0.005 | 161 | 500 |
| STL10 | 0.05 | 0.001 | 0.01 | 161 | 500 |

# E   Additional Plots

## E.1   Accuracy on PreResNet20

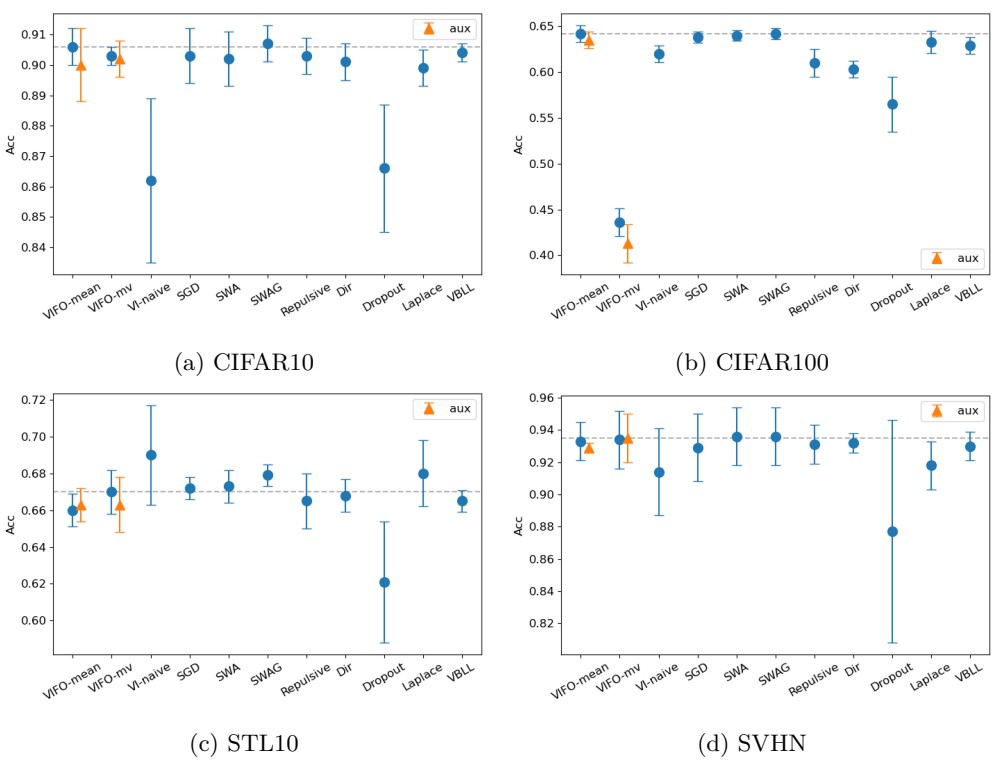

(a) CIFAR10

(b) CIFAR100

(c) STL10

(d) SVHN

Figure E.1: Test accuracy (↑) on PreResNet20. Dashed lines indicate the best version of VIFO.

## E.2 Results on AlexNet

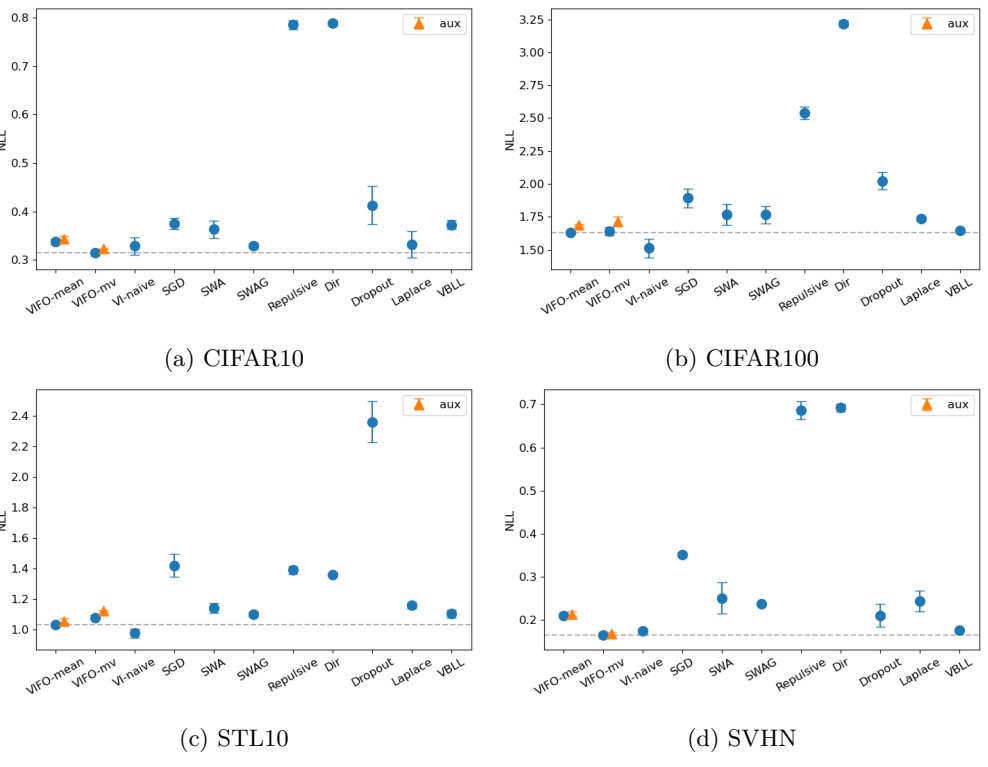

(a) CIFAR10

(b) CIFAR100

(c) STL10

(d) SVHN

Figure E.2: Test log loss (↓) of image datasets on AlexNet. Dashed lines indicate the best version of VIFO. The error bar is three times of the standard deviation for better visualization and same for other figures.

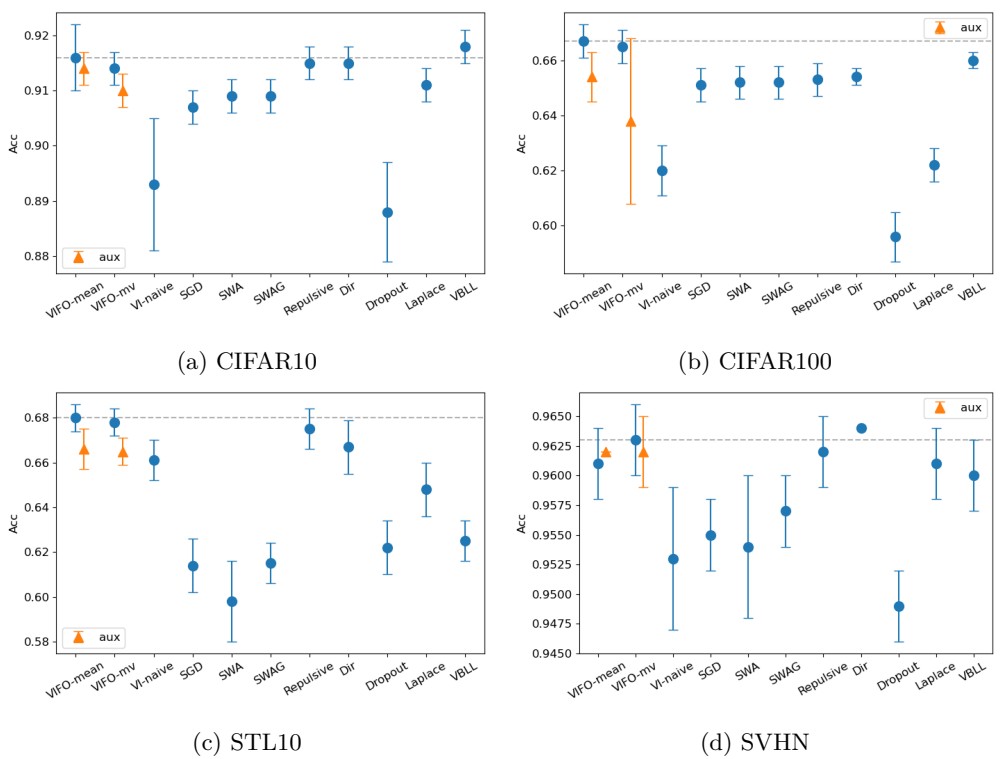

(a) CIFAR10

(b) CIFAR100

(c) STL10

(d) SVHN

Figure E.3: Test accuracy (↑) on AlexNet. Dashed lines indicate the best version of VIFO.

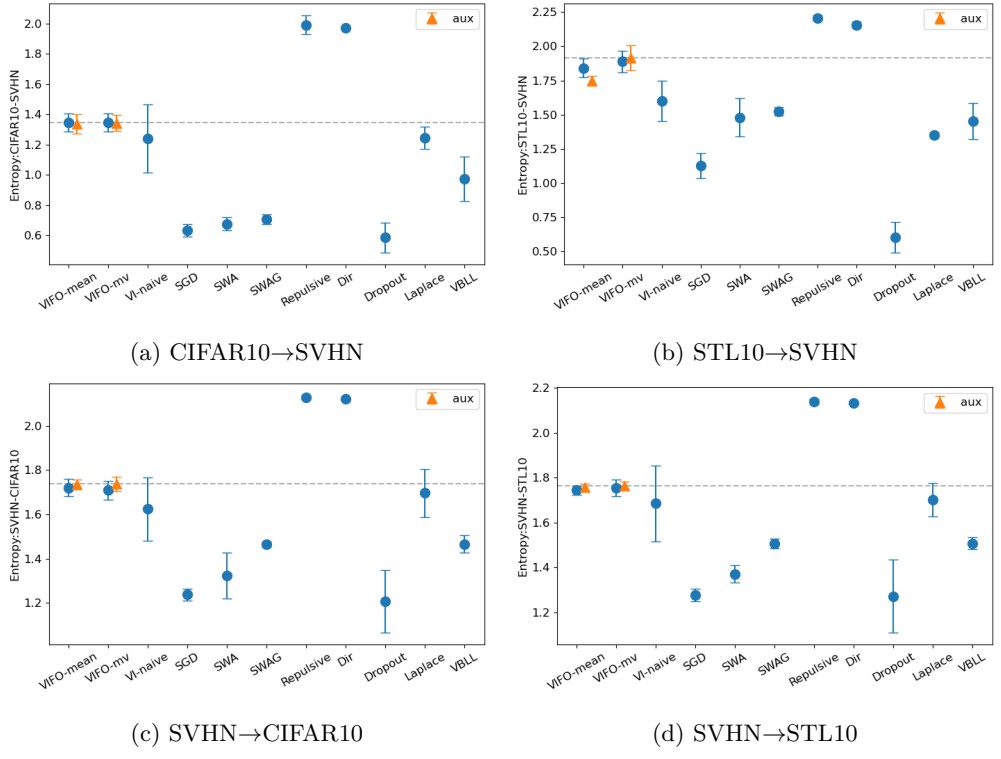

(a) CIFAR10→SVHN

(b) STL10→SVHN

(c) SVHN→CIFAR10

(d) SVHN→STL10

Figure E.4: Entropy (↑) on AlexNet.

## E.3    AUROC Comparisons

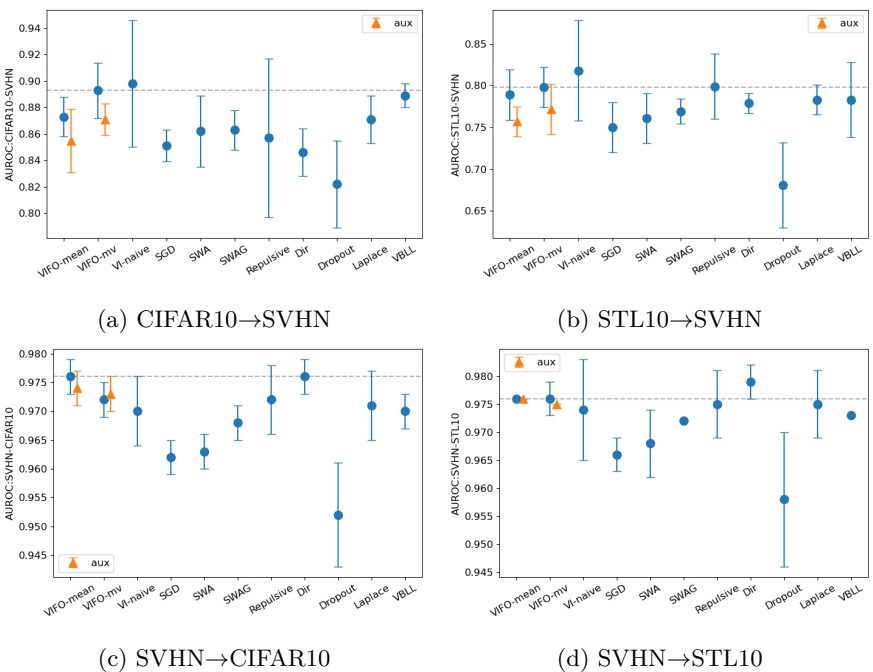

(a) CIFAR10→SVHN

(b) STL10→SVHN

(c) SVHN→CIFAR10

(d) SVHN→STL10

Figure E.5: AUROC (↑) on AlexNet.

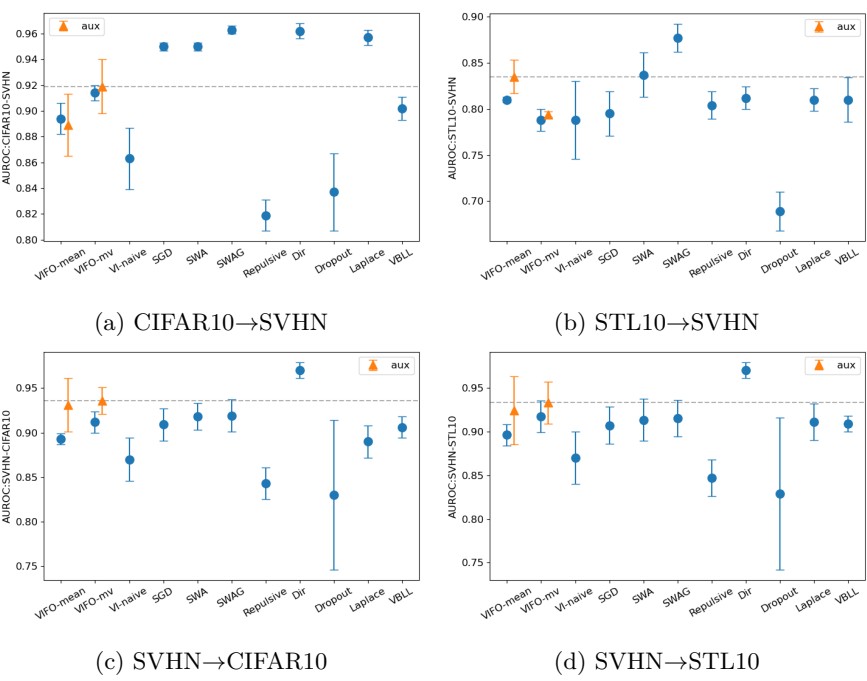

(a) CIFAR10→SVHN

(b) STL10→SVHN

(c) SVHN→CIFAR10

(d) SVHN→STL10

Figure E.6: AUROC (↑) on PreResNet20.

### E.4 Learning Curves

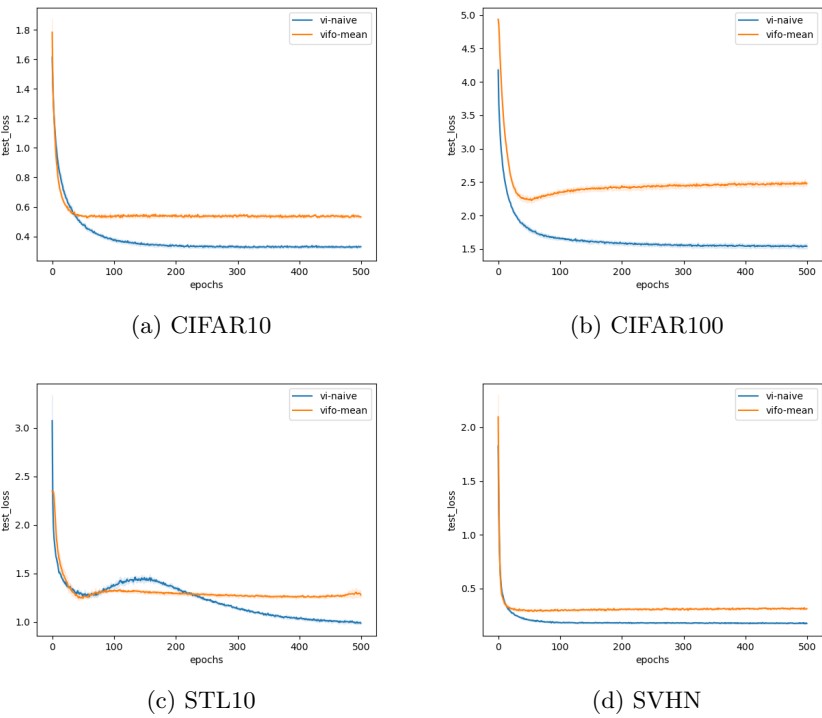

(a) CIFAR10

(b) CIFAR100

(c) STL10

(d) SVHN

Figure E.7: Learning curves for all datasets on AlexNet. We conducted 5 independent runs and report the mean and standard deviation (which is very small). The results show that in all cases, VIFO-mean converges as fast as, or faster than, VI.

**E.5   Ensembles**

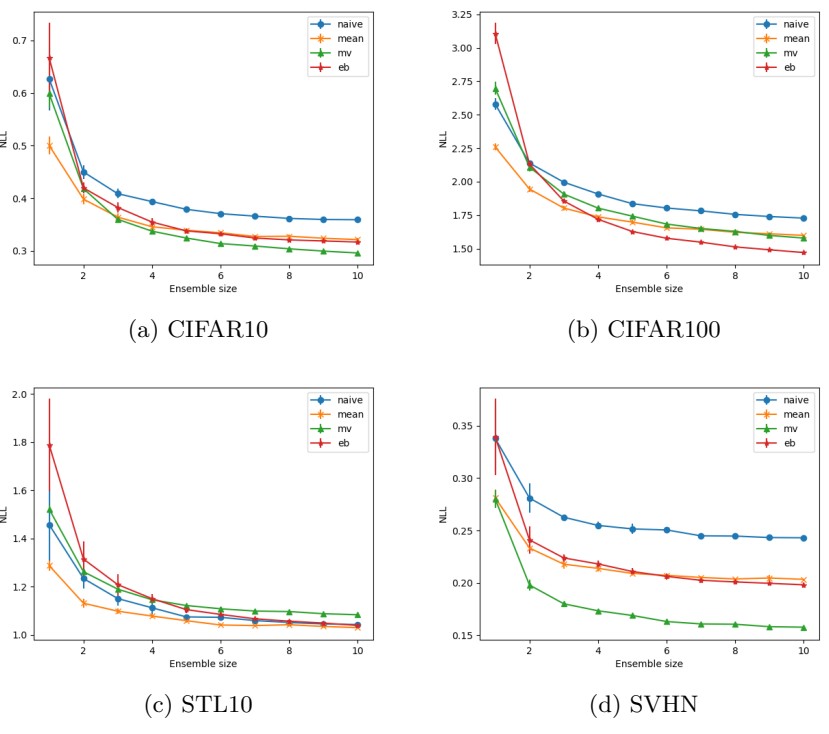

(a) CIFAR10

(b) CIFAR100

(c) STL10

(d) SVHN

Figure E.8: Test losses vs. size of ensemble. Results are shown for 5 independent runs on AlexNet.

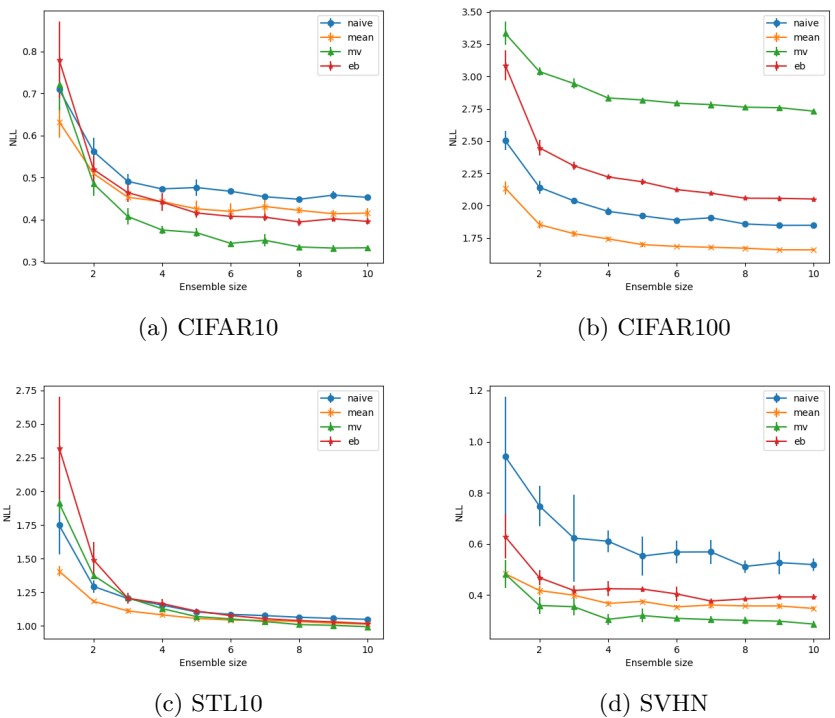

(a) CIFAR10

(b) CIFAR100

(c) STL10

(d) SVHN

Figure E.9: Test losses vs. size of ensemble. Results are shown for 5 independent runs on PreResNet20.

# F    Numerical Results

Table F.1: CIFAR10, AlexNet, NLL

| method | $\eta_{\mathrm{aux}} = 0$ | $\eta_{\mathrm{aux}} = 0.1$ | $\eta_{\mathrm{aux}} = 0.5$ | $\eta_{\mathrm{aux}} = 1.0$ |
|---|---|---|---|---|
| VIFO-naive | $0.388 \pm 0.005$ | $0.383 \pm 0.002$ | $0.382 \pm 0.001$ | $0.383 \pm 0.002$ |
| VIFO-mean | $0.338 \pm 0.002$ | $0.343 \pm 0.002$ | $0.344 \pm 0.003$ | $0.345 \pm 0.004$ |
| VIFO-mv | $0.315 \pm 0.002$ | $0.324 \pm 0.000$ | $0.315 \pm 0.004$ | $0.311 \pm 0.002$ |
| VIFO-eb | $0.347 \pm 0.003$ | $0.345 \pm 0.001$ | $0.345 \pm 0.003$ | $0.347 \pm 0.002$ |
| VI-naive | $0.329 \pm 0.006$ | | | |
| VI-mean | $0.350 \pm 0.010$ | | | |
| VI-mv | $0.315 \pm 0.013$ | | | |
| VI-eb | $0.343 \pm 0.004$ | | | |
| SGD | $0.375 \pm 0.004$ | | | |
| SWA | $0.363 \pm 0.006$ | | | |
| SWAG | $0.329 \pm 0.002$ | | | |
| Repulsive | $0.785 \pm 0.003$ | | | |
| Dir | $0.788 \pm 0.002$ | | | |
| Dropout | $0.413 \pm 0.013$ | | | |
| Laplace | $0.332 \pm 0.009$ | | | |
| VBLL | $0.373 \pm 0.003$ | | | |

Table F.2: CIFAR100, AlexNet, NLL

| method | $\eta_{\mathrm{aux}} = 0$ | $\eta_{\mathrm{aux}} = 0.1$ | $\eta_{\mathrm{aux}} = 0.5$ | $\eta_{\mathrm{aux}} = 1.0$ |
|---|---|---|---|---|
| VIFO-naive | $1.774 \pm 0.008$ | $1.840 \pm 0.016$ | $1.784 \pm 0.008$ | $1.811 \pm 0.012$ |
| VIFO-mean | $1.632 \pm 0.003$ | $1.687 \pm 0.002$ | $1.682 \pm 0.017$ | $1.683 \pm 0.021$ |
| VIFO-mv | $1.642 \pm 0.011$ | $1.716 \pm 0.012$ | $1.971 \pm 0.053$ | $2.250 \pm 0.105$ |
| VIFO-eb | $1.643 \pm 0.008$ | $1.651 \pm 0.003$ | $1.643 \pm 0.004$ | $1.649 \pm 0.006$ |
| VI-naive | $1.513 \pm 0.024$ | | | |
| VI-mean | $1.642 \pm 0.023$ | | | |
| VI-mv | $1.817 \pm 0.022$ | | | |
| VI-eb | $1.441 \pm 0.016$ | | | |
| SGD | $1.894 \pm 0.024$ | | | |
| SWA | $1.768 \pm 0.026$ | | | |
| SWAG | $1.768 \pm 0.022$ | | | |
| Repulsive | $2.540 \pm 0.016$ | | | |
| Dir | $3.218 \pm 0.008$ | | | |
| Dropout | $2.024 \pm 0.022$ | | | |
| Laplace | $1.738 \pm 0.006$ | | | |
| VBLL | $1.649 \pm 0.003$ | | | |

Table F.3: STL10, AlexNet, NLL

| method | $\eta_{\mathrm{aux}} = 0$ | $\eta_{\mathrm{aux}} = 0.1$ | $\eta_{\mathrm{aux}} = 0.5$ | $\eta_{\mathrm{aux}} = 1.0$ |
|---|---|---|---|---|
| VIFO-naive | $1.113 \pm 0.006$ | $1.117 \pm 0.005$ | $1.112 \pm 0.005$ | $1.135 \pm 0.008$ |
| VIFO-mean | $1.030 \pm 0.004$ | $1.056 \pm 0.005$ | $1.066 \pm 0.005$ | $1.067 \pm 0.011$ |
| VIFO-mv | $1.078 \pm 0.005$ | $1.121 \pm 0.002$ | $1.112 \pm 0.010$ | $1.101 \pm 0.008$ |
| VIFO-eb | $1.141 \pm 0.007$ | $1.134 \pm 0.008$ | $1.127 \pm 0.007$ | $1.135 \pm 0.002$ |
| VI-naive | $0.975 \pm 0.010$ | | | |
| VI-mean | $1.021 \pm 0.013$ | | | |
| VI-mv | $1.095 \pm 0.018$ | | | |
| VI-eb | $1.560 \pm 0.054$ | | | |
| SGD | $1.419 \pm 0.025$ | | | |
| SWA | $1.139 \pm 0.011$ | | | |
| SWAG | $1.098 \pm 0.005$ | | | |
| Repulsive | $1.388 \pm 0.008$ | | | |
| Dir | $1.359 \pm 0.003$ | | | |
| Dropout | $2.359 \pm 0.045$ | | | |
| Laplace | $1.157 \pm 0.007$ | | | |
| VBLL | $1.102 \pm 0.008$ | | | |

Table F.4: SVHN, AlexNet, NLL

| method | $\eta_{\mathrm{aux}} = 0$ | $\eta_{\mathrm{aux}} = 0.1$ | $\eta_{\mathrm{aux}} = 0.5$ | $\eta_{\mathrm{aux}} = 1.0$ |
|---|---|---|---|---|
| VIFO-naive | $0.253 \pm 0.001$ | $0.256 \pm 0.001$ | $0.257 \pm 0.001$ | $0.251 \pm 0.002$ |
| VIFO-mean | $0.211 \pm 0.002$ | $0.214 \pm 0.002$ | $0.209 \pm 0.001$ | $0.208 \pm 0.001$ |
| VIFO-mv | $0.165 \pm 0.002$ | $0.169 \pm 0.001$ | $0.166 \pm 0.001$ | $0.170 \pm 0.002$ |
| VIFO-eb | $0.211 \pm 0.001$ | $0.214 \pm 0.002$ | $0.212 \pm 0.002$ | $0.214 \pm 0.001$ |
| VI-naive | $0.175 \pm 0.002$ | | | |
| VI-mean | $0.219 \pm 0.034$ | | | |
| VI-mv | $0.173 \pm 0.003$ | | | |
| VI-eb | $0.182 \pm 0.004$ | | | |
| SGD | $0.351 \pm 0.002$ | | | |
| SWA | $0.251 \pm 0.012$ | | | |
| SWAG | $0.238 \pm 0.001$ | | | |
| Repulsive | $0.686 \pm 0.007$ | | | |
| Dir | $0.692 \pm 0.003$ | | | |
| Dropout | $0.211 \pm 0.009$ | | | |
| Laplace | $0.244 \pm 0.008$ | | | |
| VBLL | $0.177 \pm 0.001$ | | | |

Table F.5: CIFAR10, PreResNet20, NLL

| method | $\eta_{\text{aux}} = 0$ | $\eta_{\text{aux}} = 0.1$ | $\eta_{\text{aux}} = 0.5$ | $\eta_{\text{aux}} = 1.0$ |
|---|---|---|---|---|
| VIFO-naive | $0.473 \pm 0.005$ | $0.487 \pm 0.011$ | $0.486 \pm 0.010$ | $0.479 \pm 0.008$ |
| VIFO-mean | $0.393 \pm 0.010$ | $0.433 \pm 0.018$ | $0.411 \pm 0.013$ | $0.430 \pm 0.020$ |
| VIFO-mv | $0.361 \pm 0.010$ | $0.371 \pm 0.010$ | $0.367 \pm 0.006$ | $0.363 \pm 0.016$ |
| VIFO-eb | $0.419 \pm 0.014$ | $0.429 \pm 0.016$ | $0.413 \pm 0.007$ | $0.425 \pm 0.008$ |
| VI-naive | $0.410 \pm 0.028$ | | | |
| VI-mean | $0.415 \pm 0.032$ | | | |
| VI-mv | $0.437 \pm 0.029$ | | | |
| VI-eb | $0.429 \pm 0.035$ | | | |
| SGD | $0.335 \pm 0.013$ | | | |
| SWA | $0.336 \pm 0.008$ | | | |
| SWAG | $0.307 \pm 0.010$ | | | |
| Repulsive | $0.875 \pm 0.007$ | | | |
| Dir | $0.961 \pm 0.022$ | | | |
| Dropout | $0.423 \pm 0.026$ | | | |
| Laplace | $0.324 \pm 0.009$ | | | |
| VBLL | $0.366 \pm 0.006$ | | | |

Table F.6: CIFAR100, PreResNet20, NLL

| method | $\eta_{\text{aux}} = 0$ | $\eta_{\text{aux}} = 0.1$ | $\eta_{\text{aux}} = 0.5$ | $\eta_{\text{aux}} = 1.0$ |
|---|---|---|---|---|
| VIFO-naive | $1.880 \pm 0.009$ | $1.935 \pm 0.020$ | $1.864 \pm 0.008$ | $1.844 \pm 0.011$ |
| VIFO-mean | $1.632 \pm 0.013$ | $1.728 \pm 0.005$ | $1.686 \pm 0.004$ | $1.731 \pm 0.018$ |
| VIFO-mv | $2.726 \pm 0.021$ | $2.826 \pm 0.008$ | $2.899 \pm 0.019$ | $2.867 \pm 0.014$ |
| VIFO-eb | $2.076 \pm 0.006$ | $2.147 \pm 0.004$ | $2.340 \pm 0.043$ | $2.503 \pm 0.034$ |
| VI-naive | $1.642 \pm 0.030$ | | | |
| VI-mean | $1.753 \pm 0.086$ | | | |
| VI-mv | $1.804 \pm 0.089$ | | | |
| VI-eb | $1.731 \pm 0.097$ | | | |
| SGD | $1.445 \pm 0.021$ | | | |
| SWA | $1.355 \pm 0.023$ | | | |
| SWAG | $1.354 \pm 0.019$ | | | |
| Repulsive | $2.948 \pm 0.020$ | | | |
| Dir | $3.580 \pm 0.009$ | | | |
| Dropout | $1.644 \pm 0.045$ | | | |
| Laplace | $1.550 \pm 0.014$ | | | |
| VBLL | $1.484 \pm 0.017$ | | | |

Table F.7: STL10, PreResNet20, NLL

| method | $\eta_{\text{aux}} = 0$ | $\eta_{\text{aux}} = 0.1$ | $\eta_{\text{aux}} = 0.5$ | $\eta_{\text{aux}} = 1.0$ |
|---|---|---|---|---|
| VIFO-naive | $1.146 \pm 0.012$ | $1.159 \pm 0.008$ | $1.155 \pm 0.015$ | $1.145 \pm 0.005$ |
| VIFO-mean | $1.067 \pm 0.009$ | $1.066 \pm 0.011$ | $1.056 \pm 0.007$ | $1.069 \pm 0.009$ |
| VIFO-mv | $1.070 \pm 0.017$ | $1.078 \pm 0.003$ | $1.067 \pm 0.019$ | $1.073 \pm 0.011$ |
| VIFO-eb | $1.162 \pm 0.015$ | $1.164 \pm 0.020$ | $1.191 \pm 0.014$ | $1.180 \pm 0.016$ |
| VI-naive | $0.920 \pm 0.032$ | | | |
| VI-mean | $1.000 \pm 0.029$ | | | |
| VI-mv | $1.002 \pm 0.036$ | | | |
| VI-eb | $1.018 \pm 0.028$ | | | |
| SGD | $1.203 \pm 0.008$ | | | |
| SWA | $1.100 \pm 0.006$ | | | |
| SWAG | $1.100 \pm 0.010$ | | | |
| Repulsive | $1.365 \pm 0.009$ | | | |
| Dir | $1.418 \pm 0.019$ | | | |
| Dropout | $1.665 \pm 0.059$ | | | |
| Laplace | $1.130 \pm 0.011$ | | | |
| VBLL | $1.072 \pm 0.009$ | | | |

Table F.8: SVHN, PreResNet20, NLL

| method | $\eta_{\text{aux}} = 0$ | $\eta_{\text{aux}} = 0.1$ | $\eta_{\text{aux}} = 0.5$ | $\eta_{\text{aux}} = 1.0$ |
|---|---|---|---|---|
| VIFO-naive | $0.391 \pm 0.019$ | $0.603 \pm 0.057$ | $0.486 \pm 0.050$ | $0.479 \pm 0.018$ |
| VIFO-mean | $0.341 \pm 0.004$ | $0.357 \pm 0.004$ | $0.385 \pm 0.015$ | $0.334 \pm 0.015$ |
| VIFO-mv | $0.269 \pm 0.009$ | $0.297 \pm 0.010$ | $0.323 \pm 0.013$ | $0.314 \pm 0.009$ |
| VIFO-eb | $0.359 \pm 0.002$ | $0.391 \pm 0.008$ | $0.452 \pm 0.018$ | $0.402 \pm 0.028$ |
| VI-naive | $0.314 \pm 0.024$ | | | |
| VI-mean | $0.342 \pm 0.040$ | | | |
| VI-mv | $0.359 \pm 0.033$ | | | |
| VI-eb | $0.379 \pm 0.057$ | | | |
| SGD | $0.337 \pm 0.011$ | | | |
| SWA | $0.320 \pm 0.009$ | | | |
| SWAG | $0.320 \pm 0.009$ | | | |
| Repulsive | $0.822 \pm 0.020$ | | | |
| Dir | $0.845 \pm 0.031$ | | | |
| Dropout | $0.421 \pm 0.068$ | | | |
| Laplace | $0.344 \pm 0.017$ | | | |
| VBLL | $0.324 \pm 0.008$ | | | |

Table F.9: CIFAR10, AlexNet, Accuracy

| method | $\eta_{\text{aux}} = 0$ | $\eta_{\text{aux}} = 0.1$ | $\eta_{\text{aux}} = 0.5$ | $\eta_{\text{aux}} = 1.0$ |
|---|---|---|---|---|
| VIFO-naive | $0.914 \pm 0.001$ | $0.916 \pm 0.001$ | $0.915 \pm 0.001$ | $0.916 \pm 0.001$ |
| VIFO-mean | $0.916 \pm 0.002$ | $0.914 \pm 0.001$ | $0.914 \pm 0.001$ | $0.912 \pm 0.001$ |
| VIFO-mv | $0.914 \pm 0.001$ | $0.910 \pm 0.001$ | $0.912 \pm 0.002$ | $0.915 \pm 0.002$ |
| VIFO-eb | $0.914 \pm 0.002$ | $0.914 \pm 0.001$ | $0.913 \pm 0.001$ | $0.914 \pm 0.002$ |
| VI-naive | $0.893 \pm 0.004$ | | | |
| VI-mean | $0.884 \pm 0.004$ | | | |
| VI-mv | $0.901 \pm 0.003$ | | | |
| VI-eb | $0.886 \pm 0.003$ | | | |
| SGD | $0.907 \pm 0.001$ | | | |
| SWA | $0.909 \pm 0.001$ | | | |
| SWAG | $0.909 \pm 0.001$ | | | |
| Repulsive | $0.915 \pm 0.001$ | | | |
| Dir | $0.915 \pm 0.001$ | | | |
| Dropout | $0.888 \pm 0.003$ | | | |
| Laplace | $0.911 \pm 0.001$ | | | |
| VBLL | $0.918 \pm 0.001$ | | | |

Table F.10: CIFAR100, AlexNet, Accuracy

| method | $\eta_{\text{aux}} = 0$ | $\eta_{\text{aux}} = 0.1$ | $\eta_{\text{aux}} = 0.5$ | $\eta_{\text{aux}} = 1.0$ |
|---|---|---|---|---|
| VIFO-naive | $0.658 \pm 0.003$ | $0.651 \pm 0.004$ | $0.657 \pm 0.003$ | $0.644 \pm 0.003$ |
| VIFO-mean | $0.667 \pm 0.002$ | $0.654 \pm 0.003$ | $0.650 \pm 0.007$ | $0.658 \pm 0.002$ |
| VIFO-mv | $0.665 \pm 0.002$ | $0.638 \pm 0.010$ | $0.568 \pm 0.038$ | $0.558 \pm 0.028$ |
| VIFO-eb | $0.669 \pm 0.002$ | $0.671 \pm 0.001$ | $0.668 \pm 0.003$ | $0.664 \pm 0.002$ |
| VI-naive | $0.620 \pm 0.003$ | | | |
| VI-mean | $0.608 \pm 0.002$ | | | |
| VI-mv | $0.606 \pm 0.002$ | | | |
| VI-eb | $0.629 \pm 0.004$ | | | |
| SGD | $0.651 \pm 0.002$ | | | |
| SWA | $0.652 \pm 0.002$ | | | |
| SWAG | $0.652 \pm 0.002$ | | | |
| Repulsive | $0.653 \pm 0.002$ | | | |
| Dir | $0.654 \pm 0.001$ | | | |
| Dropout | $0.596 \pm 0.003$ | | | |
| Laplace | $0.622 \pm 0.002$ | | | |
| VBLL | $0.660 \pm 0.001$ | | | |

Table F.11: STL10, AlexNet, Accuracy

| method | $\eta_{\mathrm{aux}} = 0$ | $\eta_{\mathrm{aux}} = 0.1$ | $\eta_{\mathrm{aux}} = 0.5$ | $\eta_{\mathrm{aux}} = 1.0$ |
|---|---|---|---|---|
| VIFO-naive | $0.670 \pm 0.003$ | $0.672 \pm 0.002$ | $0.674 \pm 0.002$ | $0.665 \pm 0.001$ |
| VIFO-mean | $0.680 \pm 0.002$ | $0.666 \pm 0.003$ | $0.669 \pm 0.003$ | $0.663 \pm 0.003$ |
| VIFO-mv | $0.678 \pm 0.002$ | $0.665 \pm 0.002$ | $0.665 \pm 0.002$ | $0.668 \pm 0.002$ |
| VIFO-eb | $0.662 \pm 0.002$ | $0.668 \pm 0.002$ | $0.670 \pm 0.004$ | $0.670 \pm 0.002$ |
| VI-naive | $0.661 \pm 0.003$ | | | |
| VI-mean | $0.649 \pm 0.009$ | | | |
| VI-mv | $0.644 \pm 0.006$ | | | |
| VI-eb | $0.414 \pm 0.025$ | | | |
| SGD | $0.614 \pm 0.004$ | | | |
| SWA | $0.598 \pm 0.006$ | | | |
| SWAG | $0.615 \pm 0.003$ | | | |
| Repulsive | $0.675 \pm 0.003$ | | | |
| Dir | $0.667 \pm 0.004$ | | | |
| Dropout | $0.622 \pm 0.004$ | | | |
| Laplace | $0.648 \pm 0.004$ | | | |
| VBLL | $0.625 \pm 0.003$ | | | |

Table F.12: SVHN, AlexNet, Accuracy

| method | $\eta_{\mathrm{aux}} = 0$ | $\eta_{\mathrm{aux}} = 0.1$ | $\eta_{\mathrm{aux}} = 0.5$ | $\eta_{\mathrm{aux}} = 1.0$ |
|---|---|---|---|---|
| VIFO-naive | $0.962 \pm 0.001$ | $0.962 \pm 0.001$ | $0.962 \pm 0.000$ | $0.962 \pm 0.001$ |
| VIFO-mean | $0.961 \pm 0.001$ | $0.962 \pm 0.000$ | $0.962 \pm 0.001$ | $0.963 \pm 0.001$ |
| VIFO-mv | $0.963 \pm 0.001$ | $0.962 \pm 0.001$ | $0.963 \pm 0.000$ | $0.962 \pm 0.000$ |
| VIFO-eb | $0.962 \pm 0.000$ | $0.961 \pm 0.001$ | $0.961 \pm 0.000$ | $0.961 \pm 0.000$ |
| VI-naive | $0.953 \pm 0.002$ | | | |
| VI-mean | $0.945 \pm 0.008$ | | | |
| VI-mv | $0.955 \pm 0.001$ | | | |
| VI-eb | $0.950 \pm 0.001$ | | | |
| SGD | $0.955 \pm 0.001$ | | | |
| SWA | $0.954 \pm 0.002$ | | | |
| SWAG | $0.957 \pm 0.001$ | | | |
| Repulsive | $0.962 \pm 0.001$ | | | |
| Dir | $0.964 \pm 0.000$ | | | |
| Dropout | $0.949 \pm 0.001$ | | | |
| Laplace | $0.961 \pm 0.001$ | | | |
| VBLL | $0.960 \pm 0.001$ | | | |

Table F.13: CIFAR10, PreResNet20, Accuracy

| method | $\eta_{\text{aux}} = 0$ | $\eta_{\text{aux}} = 0.1$ | $\eta_{\text{aux}} = 0.5$ | $\eta_{\text{aux}} = 1.0$ |
|---|---|---|---|---|
| VIFO-naive | $0.897 \pm 0.003$ | $0.896 \pm 0.002$ | $0.893 \pm 0.003$ | $0.899 \pm 0.003$ |
| VIFO-mean | $0.906 \pm 0.002$ | $0.900 \pm 0.004$ | $0.907 \pm 0.002$ | $0.904 \pm 0.005$ |
| VIFO-mv | $0.903 \pm 0.001$ | $0.902 \pm 0.002$ | $0.902 \pm 0.001$ | $0.903 \pm 0.002$ |
| VIFO-eb | $0.900 \pm 0.003$ | $0.900 \pm 0.005$ | $0.902 \pm 0.002$ | $0.901 \pm 0.003$ |
| VI-naive | $0.862 \pm 0.009$ | | | |
| VI-mean | $0.867 \pm 0.009$ | | | |
| VI-mv | $0.864 \pm 0.010$ | | | |
| VI-eb | $0.859 \pm 0.011$ | | | |
| SGD | $0.903 \pm 0.003$ | | | |
| SWA | $0.902 \pm 0.003$ | | | |
| SWAG | $0.907 \pm 0.002$ | | | |
| Repulsive | $0.903 \pm 0.002$ | | | |
| Dir | $0.901 \pm 0.002$ | | | |
| Dropout | $0.866 \pm 0.007$ | | | |
| Laplace | $0.899 \pm 0.002$ | | | |
| VBLL | $0.904 \pm 0.001$ | | | |

Table F.14: CIFAR100, PreResNet20, Accuracy

| method | $\eta_{\text{aux}} = 0$ | $\eta_{\text{aux}} = 0.1$ | $\eta_{\text{aux}} = 0.5$ | $\eta_{\text{aux}} = 1.0$ |
|---|---|---|---|---|
| VIFO-naive | $0.631 \pm 0.004$ | $0.620 \pm 0.005$ | $0.631 \pm 0.002$ | $0.628 \pm 0.002$ |
| VIFO-mean | $0.642 \pm 0.003$ | $0.635 \pm 0.003$ | $0.643 \pm 0.003$ | $0.637 \pm 0.006$ |
| VIFO-mv | $0.436 \pm 0.005$ | $0.413 \pm 0.007$ | $0.398 \pm 0.006$ | $0.405 \pm 0.004$ |
| VIFO-eb | $0.579 \pm 0.004$ | $0.567 \pm 0.009$ | $0.535 \pm 0.004$ | $0.499 \pm 0.017$ |
| VI-naive | $0.620 \pm 0.003$ | | | |
| VI-mean | $0.608 \pm 0.002$ | | | |
| VI-mv | $0.606 \pm 0.002$ | | | |
| VI-eb | $0.629 \pm 0.004$ | | | |
| SGD | $0.638 \pm 0.002$ | | | |
| SWA | $0.640 \pm 0.002$ | | | |
| SWAG | $0.642 \pm 0.002$ | | | |
| Repulsive | $0.610 \pm 0.005$ | | | |
| Dir | $0.603 \pm 0.003$ | | | |
| Dropout | $0.565 \pm 0.010$ | | | |
| Laplace | $0.633 \pm 0.004$ | | | |
| VBLL | $0.629 \pm 0.003$ | | | |

Table F.15: STL10, PreResNet20, Accuracy

| method | $\eta_{\text{aux}} = 0$ | $\eta_{\text{aux}} = 0.1$ | $\eta_{\text{aux}} = 0.5$ | $\eta_{\text{aux}} = 1.0$ |
|---|---|---|---|---|
| VIFO-naive | $0.658 \pm 0.004$ | $0.658 \pm 0.002$ | $0.656 \pm 0.001$ | $0.656 \pm 0.003$ |
| VIFO-mean | $0.660 \pm 0.003$ | $0.663 \pm 0.003$ | $0.669 \pm 0.004$ | $0.663 \pm 0.004$ |
| VIFO-mv | $0.670 \pm 0.004$ | $0.663 \pm 0.005$ | $0.662 \pm 0.005$ | $0.662 \pm 0.003$ |
| VIFO-eb | $0.661 \pm 0.004$ | $0.661 \pm 0.004$ | $0.657 \pm 0.003$ | $0.660 \pm 0.005$ |
| VI-naive | $0.690 \pm 0.009$ | | | |
| VI-mean | $0.674 \pm 0.010$ | | | |
| VI-mv | $0.688 \pm 0.009$ | | | |
| VI-eb | $0.638 \pm 0.011$ | | | |
| SGD | $0.672 \pm 0.002$ | | | |
| SWA | $0.673 \pm 0.003$ | | | |
| SWAG | $0.679 \pm 0.002$ | | | |
| Repulsive | $0.665 \pm 0.005$ | | | |
| Dir | $0.668 \pm 0.003$ | | | |
| Dropout | $0.621 \pm 0.011$ | | | |
| Laplace | $0.680 \pm 0.006$ | | | |
| VBLL | $0.665 \pm 0.002$ | | | |

Table F.16: SVHN, PreResNet20, Accuracy

| method | $\eta_{\text{aux}} = 0$ | $\eta_{\text{aux}} = 0.1$ | $\eta_{\text{aux}} = 0.5$ | $\eta_{\text{aux}} = 1.0$ |
|---|---|---|---|---|
| VIFO-naive | $0.930 \pm 0.004$ | $0.911 \pm 0.014$ | $0.912 \pm 0.004$ | $0.915 \pm 0.008$ |
| VIFO-mean | $0.933 \pm 0.004$ | $0.929 \pm 0.001$ | $0.930 \pm 0.001$ | $0.936 \pm 0.009$ |
| VIFO-mv | $0.934 \pm 0.006$ | $0.935 \pm 0.005$ | $0.928 \pm 0.008$ | $0.930 \pm 0.002$ |
| VIFO-eb | $0.929 \pm 0.004$ | $0.917 \pm 0.005$ | $0.905 \pm 0.003$ | $0.924 \pm 0.007$ |
| VI-naive | $0.914 \pm 0.009$ | | | |
| VI-mean | $0.901 \pm 0.015$ | | | |
| VI-mv | $0.902 \pm 0.012$ | | | |
| VI-eb | $0.890 \pm 0.019$ | | | |
| SGD | $0.929 \pm 0.007$ | | | |
| SWA | $0.936 \pm 0.006$ | | | |
| SWAG | $0.936 \pm 0.006$ | | | |
| Repulsive | $0.931 \pm 0.004$ | | | |
| Dir | $0.932 \pm 0.002$ | | | |
| Dropout | $0.877 \pm 0.023$ | | | |
| Laplace | $0.918 \pm 0.005$ | | | |
| VBLL | $0.930 \pm 0.003$ | | | |

Table F.17: ECE:CIFAR10-STL10, AlexNet

| method | $\eta_{\text{aux}} = 0$ | $\eta_{\text{aux}} = 0.1$ | $\eta_{\text{aux}} = 0.5$ | $\eta_{\text{aux}} = 1.0$ |
|---|---|---|---|---|
| VIFO-naive | $0.042 \pm 0.003$ | $0.042 \pm 0.002$ | $0.047 \pm 0.004$ | $0.045 \pm 0.003$ |
| VIFO-mean | $0.056 \pm 0.002$ | $0.053 \pm 0.003$ | $0.055 \pm 0.001$ | $0.059 \pm 0.002$ |
| VIFO-mv | $0.067 \pm 0.002$ | $0.069 \pm 0.002$ | $0.068 \pm 0.002$ | $0.068 \pm 0.003$ |
| VIFO-eb | $0.039 \pm 0.003$ | $0.042 \pm 0.002$ | $0.038 \pm 0.002$ | $0.038 \pm 0.002$ |
| VI-naive | $0.108 \pm 0.002$ | | | |
| VI-mean | $0.105 \pm 0.007$ | | | |
| VI-mv | $0.131 \pm 0.003$ | | | |
| VI-eb | $0.118 \pm 0.004$ | | | |
| SGD | $0.152 \pm 0.002$ | | | |
| SWA | $0.150 \pm 0.001$ | | | |
| SWAG | $0.144 \pm 0.001$ | | | |
| Repulsive | $0.246 \pm 0.003$ | | | |
| Dir | $0.248 \pm 0.002$ | | | |
| Dropout | $0.180 \pm 0.002$ | | | |
| Laplace | $0.169 \pm 0.007$ | | | |
| VBLL | $0.095 \pm 0.002$ | | | |

Table F.18: ECE:STL10-CIFAR10, AlexNet

| method | $\eta_{\text{aux}} = 0$ | $\eta_{\text{aux}} = 0.1$ | $\eta_{\text{aux}} = 0.5$ | $\eta_{\text{aux}} = 1.0$ |
|---|---|---|---|---|
| VIFO-naive | $0.053 \pm 0.004$ | $0.053 \pm 0.004$ | $0.040 \pm 0.004$ | $0.034 \pm 0.004$ |
| VIFO-mean | $0.065 \pm 0.002$ | $0.063 \pm 0.003$ | $0.055 \pm 0.003$ | $0.046 \pm 0.002$ |
| VIFO-mv | $0.075 \pm 0.002$ | $0.067 \pm 0.003$ | $0.078 \pm 0.003$ | $0.073 \pm 0.002$ |
| VIFO-eb | $0.081 \pm 0.004$ | $0.078 \pm 0.002$ | $0.066 \pm 0.002$ | $0.075 \pm 0.004$ |
| VI-naive | $0.072 \pm 0.007$ | | | |
| VI-mean | $0.107 \pm 0.007$ | | | |
| VI-mv | $0.155 \pm 0.004$ | | | |
| VI-eb | $0.030 \pm 0.006$ | | | |
| SGD | $0.238 \pm 0.007$ | | | |
| SWA | $0.137 \pm 0.016$ | | | |
| SWAG | $0.117 \pm 0.007$ | | | |
| Repulsive | $0.197 \pm 0.006$ | | | |
| Dir | $0.146 \pm 0.004$ | | | |
| Dropout | $0.365 \pm 0.007$ | | | |
| Laplace | $0.085 \pm 0.004$ | | | |
| VBLL | $0.129 \pm 0.007$ | | | |

Table F.19: ECE:CIFAR10-STL10, PreResNet20

| method | $\eta_{\text{aux}} = 0$ | $\eta_{\text{aux}} = 0.1$ | $\eta_{\text{aux}} = 0.5$ | $\eta_{\text{aux}} = 1.0$ |
|---|---|---|---|---|
| VIFO-naive | $0.029 \pm 0.001$ | $0.032 \pm 0.003$ | $0.032 \pm 0.004$ | $0.037 \pm 0.004$ |
| VIFO-mean | $0.036 \pm 0.003$ | $0.028 \pm 0.005$ | $0.032 \pm 0.002$ | $0.029 \pm 0.003$ |
| VIFO-mv | $0.058 \pm 0.003$ | $0.047 \pm 0.004$ | $0.052 \pm 0.002$ | $0.051 \pm 0.002$ |
| VIFO-eb | $0.024 \pm 0.003$ | $0.028 \pm 0.005$ | $0.031 \pm 0.002$ | $0.026 \pm 0.002$ |
| VI-naive | $0.126 \pm 0.004$ | | | |
| VI-mean | $0.147 \pm 0.002$ | | | |
| VI-mv | $0.159 \pm 0.003$ | | | |
| VI-eb | $0.137 \pm 0.005$ | | | |
| SGD | $0.092 \pm 0.004$ | | | |
| SWA | $0.089 \pm 0.001$ | | | |
| SWAG | $0.078 \pm 0.003$ | | | |
| Repulsive | $0.259 \pm 0.004$ | | | |
| Dir | $0.319 \pm 0.007$ | | | |
| Dropout | $0.146 \pm 0.002$ | | | |
| Laplace | $0.063 \pm 0.002$ | | | |
| VBLL | $0.095 \pm 0.001$ | | | |

Table F.20: ECE:STL10-CIFAR10, PreResNet20

| method | $\eta_{\text{aux}} = 0$ | $\eta_{\text{aux}} = 0.1$ | $\eta_{\text{aux}} = 0.5$ | $\eta_{\text{aux}} = 1.0$ |
|---|---|---|---|---|
| VIFO-naive | $0.021 \pm 0.001$ | $0.022 \pm 0.002$ | $0.025 \pm 0.003$ | $0.028 \pm 0.004$ |
| VIFO-mean | $0.029 \pm 0.003$ | $0.024 \pm 0.005$ | $0.019 \pm 0.002$ | $0.019 \pm 0.003$ |
| VIFO-mv | $0.088 \pm 0.005$ | $0.081 \pm 0.003$ | $0.081 \pm 0.004$ | $0.086 \pm 0.002$ |
| VIFO-eb | $0.051 \pm 0.005$ | $0.039 \pm 0.002$ | $0.030 \pm 0.003$ | $0.038 \pm 0.002$ |
| VI-naive | $0.129 \pm 0.003$ | | | |
| VI-mean | $0.151 \pm 0.004$ | | | |
| VI-mv | $0.171 \pm 0.011$ | | | |
| VI-eb | $0.094 \pm 0.005$ | | | |
| SGD | $0.112 \pm 0.002$ | | | |
| SWA | $0.107 \pm 0.003$ | | | |
| SWAG | $0.093 \pm 0.004$ | | | |
| Repulsive | $0.185 \pm 0.006$ | | | |
| Dir | $0.200 \pm 0.005$ | | | |
| Dropout | $0.290 \pm 0.007$ | | | |
| Laplace | $0.107 \pm 0.003$ | | | |
| VBLL | $0.097 \pm 0.009$ | | | |

Table F.21: Entropy:CIFAR10-SVHN, AlexNet

| method | $\eta_{\mathrm{aux}} = 0$ | $\eta_{\mathrm{aux}} = 0.1$ | $\eta_{\mathrm{aux}} = 0.5$ | $\eta_{\mathrm{aux}} = 1.0$ |
|---|---|---|---|---|
| VIFO-naive | $1.357 \pm 0.009$ | $1.399 \pm 0.023$ | $1.286 \pm 0.049$ | $1.355 \pm 0.033$ |
| VIFO-mean | $1.344 \pm 0.020$ | $1.336 \pm 0.021$ | $1.313 \pm 0.042$ | $1.342 \pm 0.039$ |
| VIFO-mv | $1.344 \pm 0.020$ | $1.342 \pm 0.018$ | $1.354 \pm 0.023$ | $1.332 \pm 0.017$ |
| VIFO-eb | $1.330 \pm 0.045$ | $1.330 \pm 0.026$ | $1.323 \pm 0.017$ | $1.296 \pm 0.027$ |
| VI-naive | $1.238 \pm 0.075$ | | | |
| VI-mean | $1.280 \pm 0.157$ | | | |
| VI-mv | $1.053 \pm 0.068$ | | | |
| VI-eb | $1.133 \pm 0.069$ | | | |
| SGD | $0.633 \pm 0.014$ | | | |
| SWA | $0.676 \pm 0.014$ | | | |
| SWAG | $0.705 \pm 0.011$ | | | |
| Repulsive | $1.990 \pm 0.021$ | | | |
| Dir | $1.970 \pm 0.002$ | | | |
| Dropout | $0.585 \pm 0.033$ | | | |
| Laplace | $1.245 \pm 0.025$ | | | |
| VBLL | $0.971 \pm 0.049$ | | | |

Table F.22: Entropy:STL10-SVHN, AlexNet

| method | $\eta_{\mathrm{aux}} = 0$ | $\eta_{\mathrm{aux}} = 0.1$ | $\eta_{\mathrm{aux}} = 0.5$ | $\eta_{\mathrm{aux}} = 1.0$ |
|---|---|---|---|---|
| VIFO-naive | $1.772 \pm 0.017$ | $1.773 \pm 0.015$ | $1.779 \pm 0.024$ | $1.763 \pm 0.011$ |
| VIFO-mean | $1.840 \pm 0.023$ | $1.748 \pm 0.012$ | $1.818 \pm 0.029$ | $1.826 \pm 0.015$ |
| VIFO-mv | $1.889 \pm 0.026$ | $1.915 \pm 0.031$ | $1.848 \pm 0.051$ | $1.859 \pm 0.018$ |
| VIFO-eb | $1.764 \pm 0.026$ | $1.754 \pm 0.018$ | $1.798 \pm 0.028$ | $1.716 \pm 0.012$ |
| VI-naive | $1.601 \pm 0.049$ | | | |
| VI-mean | $1.495 \pm 0.018$ | | | |
| VI-mv | $1.345 \pm 0.046$ | | | |
| VI-eb | $2.024 \pm 0.019$ | | | |
| SGD | $1.127 \pm 0.030$ | | | |
| SWA | $1.479 \pm 0.047$ | | | |
| SWAG | $1.525 \pm 0.010$ | | | |
| Repulsive | $2.208 \pm 0.006$ | | | |
| Dir | $2.157 \pm 0.008$ | | | |
| Dropout | $0.601 \pm 0.037$ | | | |
| Laplace | $1.349 \pm 0.007$ | | | |
| VBLL | $1.454 \pm 0.044$ | | | |

Table F.23: Entropy:SVHN-CIFAR10, AlexNet

| method | $\eta_{\mathrm{aux}} = 0$ | $\eta_{\mathrm{aux}} = 0.1$ | $\eta_{\mathrm{aux}} = 0.5$ | $\eta_{\mathrm{aux}} = 1.0$ |
|---|---|---|---|---|
| VIFO-naive | $1.694 \pm 0.008$ | $1.711 \pm 0.005$ | $1.694 \pm 0.003$ | $1.693 \pm 0.002$ |
| VIFO-mean | $1.721 \pm 0.013$ | $1.735 \pm 0.007$ | $1.729 \pm 0.009$ | $1.742 \pm 0.008$ |
| VIFO-mv | $1.709 \pm 0.014$ | $1.738 \pm 0.011$ | $1.754 \pm 0.014$ | $1.758 \pm 0.011$ |
| VIFO-eb | $1.740 \pm 0.015$ | $1.749 \pm 0.005$ | $1.739 \pm 0.007$ | $1.773 \pm 0.033$ |
| VI-naive | $1.624 \pm 0.048$ | | | |
| VI-mean | $1.711 \pm 0.052$ | | | |
| VI-mv | $1.448 \pm 0.036$ | | | |
| VI-eb | $1.577 \pm 0.058$ | | | |
| SGD | $1.237 \pm 0.009$ | | | |
| SWA | $1.323 \pm 0.035$ | | | |
| SWAG | $1.465 \pm 0.004$ | | | |
| Repulsive | $2.129 \pm 0.003$ | | | |
| Dir | $2.121 \pm 0.002$ | | | |
| Dropout | $1.206 \pm 0.047$ | | | |
| Laplace | $1.696 \pm 0.036$ | | | |
| VBLL | $1.466 \pm 0.013$ | | | |

Table F.24: Entropy:SVHN-STL10, AlexNet

| method | $\eta_{\mathrm{aux}} = 0$ | $\eta_{\mathrm{aux}} = 0.1$ | $\eta_{\mathrm{aux}} = 0.5$ | $\eta_{\mathrm{aux}} = 1.0$ |
|---|---|---|---|---|
| VIFO-naive | $1.720 \pm 0.007$ | $1.734 \pm 0.005$ | $1.711 \pm 0.004$ | $1.708 \pm 0.008$ |
| VIFO-mean | $1.744 \pm 0.007$ | $1.756 \pm 0.006$ | $1.753 \pm 0.011$ | $1.756 \pm 0.007$ |
| VIFO-mv | $1.754 \pm 0.012$ | $1.763 \pm 0.006$ | $1.804 \pm 0.017$ | $1.790 \pm 0.009$ |
| VIFO-eb | $1.764 \pm 0.016$ | $1.776 \pm 0.007$ | $1.775 \pm 0.017$ | $1.765 \pm 0.030$ |
| VI-naive | $1.685 \pm 0.056$ | | | |
| VI-mean | $1.772 \pm 0.042$ | | | |
| VI-mv | $1.512 \pm 0.041$ | | | |
| VI-eb | $1.631 \pm 0.047$ | | | |
| SGD | $1.277 \pm 0.009$ | | | |
| SWA | $1.371 \pm 0.013$ | | | |
| SWAG | $1.507 \pm 0.007$ | | | |
| Repulsive | $2.138 \pm 0.004$ | | | |
| Dir | $2.132 \pm 0.002$ | | | |
| Dropout | $1.272 \pm 0.054$ | | | |
| Laplace | $1.702 \pm 0.025$ | | | |
| VBLL | $1.508 \pm 0.009$ | | | |

Table F.25: Entropy:CIFAR10-SVHN, PreResNet20

| method | $\eta_{\text{aux}} = 0$ | $\eta_{\text{aux}} = 0.1$ | $\eta_{\text{aux}} = 0.5$ | $\eta_{\text{aux}} = 1.0$ |
|---|---|---|---|---|
| VIFO-naive | $1.465 \pm 0.007$ | $1.516 \pm 0.009$ | $1.553 \pm 0.006$ | $1.540 \pm 0.004$ |
| VIFO-mean | $1.469 \pm 0.009$ | $1.553 \pm 0.018$ | $1.599 \pm 0.005$ | $1.580 \pm 0.010$ |
| VIFO-mv | $1.559 \pm 0.013$ | $1.611 \pm 0.016$ | $1.590 \pm 0.017$ | $1.543 \pm 0.015$ |
| VIFO-eb | $1.532 \pm 0.011$ | $1.540 \pm 0.013$ | $1.582 \pm 0.018$ | $1.548 \pm 0.005$ |
| VI-naive | $1.192 \pm 0.025$ | | | |
| VI-mean | $1.097 \pm 0.015$ | | | |
| VI-mv | $1.048 \pm 0.019$ | | | |
| VI-eb | $1.174 \pm 0.049$ | | | |
| SGD | $1.398 \pm 0.009$ | | | |
| SWA | $1.398 \pm 0.015$ | | | |
| SWAG | $1.586 \pm 0.016$ | | | |
| Repulsive | $1.971 \pm 0.006$ | | | |
| Dir | $2.246 \pm 0.002$ | | | |
| Dropout | $0.995 \pm 0.022$ | | | |
| Laplace | $1.523 \pm 0.010$ | | | |
| VBLL | $1.252 \pm 0.012$ | | | |

Table F.26: Entropy:STL10-SVHN, PreResNet20

| method | $\eta_{\text{aux}} = 0$ | $\eta_{\text{aux}} = 0.1$ | $\eta_{\text{aux}} = 0.5$ | $\eta_{\text{aux}} = 1.0$ |
|---|---|---|---|---|
| VIFO-naive | $1.763 \pm 0.008$ | $1.770 \pm 0.005$ | $1.742 \pm 0.009$ | $1.728 \pm 0.007$ |
| VIFO-mean | $1.796 \pm 0.006$ | $1.908 \pm 0.008$ | $1.876 \pm 0.002$ | $1.862 \pm 0.008$ |
| VIFO-mv | $1.565 \pm 0.007$ | $1.607 \pm 0.004$ | $1.588 \pm 0.005$ | $1.574 \pm 0.008$ |
| VIFO-eb | $1.603 \pm 0.005$ | $1.637 \pm 0.011$ | $1.634 \pm 0.008$ | $1.656 \pm 0.017$ |
| VI-naive | $1.394 \pm 0.020$ | | | |
| VI-mean | $1.323 \pm 0.013$ | | | |
| VI-mv | $1.267 \pm 0.030$ | | | |
| VI-eb | $1.517 \pm 0.060$ | | | |
| SGD | $1.432 \pm 0.022$ | | | |
| SWA | $1.462 \pm 0.013$ | | | |
| SWAG | $1.579 \pm 0.013$ | | | |
| Repulsive | $2.160 \pm 0.005$ | | | |
| Dir | $2.194 \pm 0.004$ | | | |
| Dropout | $0.830 \pm 0.024$ | | | |
| Laplace | $1.313 \pm 0.006$ | | | |
| VBLL | $1.481 \pm 0.008$ | | | |

Table F.27: Entropy:SVHN-CIFAR10, PreResNet20

| method | $\eta_{\mathrm{aux}} = 0$ | $\eta_{\mathrm{aux}} = 0.1$ | $\eta_{\mathrm{aux}} = 0.5$ | $\eta_{\mathrm{aux}} = 1.0$ |
|---|---|---|---|---|
| VIFO-naive | $1.531 \pm 0.008$ | $2.019 \pm 0.026$ | $1.942 \pm 0.051$ | $1.959 \pm 0.039$ |
| VIFO-mean | $1.515 \pm 0.012$ | $1.908 \pm 0.080$ | $1.850 \pm 0.065$ | $1.780 \pm 0.028$ |
| VIFO-mv | $1.568 \pm 0.014$ | $1.758 \pm 0.051$ | $1.848 \pm 0.067$ | $1.706 \pm 0.057$ |
| VIFO-eb | $1.539 \pm 0.004$ | $1.775 \pm 0.038$ | $1.898 \pm 0.022$ | $1.923 \pm 0.015$ |
| VI-naive | $1.213 \pm 0.010$ | | | |
| VI-mean | $1.169 \pm 0.017$ | | | |
| VI-mv | $1.080 \pm 0.011$ | | | |
| VI-eb | $1.219 \pm 0.029$ | | | |
| SGD | $1.384 \pm 0.005$ | | | |
| SWA | $1.403 \pm 0.002$ | | | |
| SWAG | $1.476 \pm 0.005$ | | | |
| Repulsive | $2.006 \pm 0.013$ | | | |
| Dir | $2.250 \pm 0.001$ | | | |
| Dropout | $1.059 \pm 0.019$ | | | |
| Laplace | $1.502 \pm 0.008$ | | | |
| VBLL | $1.315 \pm 0.007$ | | | |

Table F.28: Entropy:SVHN-STL10, PreResNet20

| method | $\eta_{\mathrm{aux}} = 0$ | $\eta_{\mathrm{aux}} = 0.1$ | $\eta_{\mathrm{aux}} = 0.5$ | $\eta_{\mathrm{aux}} = 1.0$ |
|---|---|---|---|---|
| VIFO-naive | $1.529 \pm 0.016$ | $1.995 \pm 0.042$ | $1.860 \pm 0.041$ | $1.885 \pm 0.054$ |
| VIFO-mean | $1.543 \pm 0.012$ | $1.837 \pm 0.070$ | $1.875 \pm 0.066$ | $1.762 \pm 0.052$ |
| VIFO-mv | $1.588 \pm 0.010$ | $1.763 \pm 0.061$ | $1.875 \pm 0.041$ | $1.671 \pm 0.039$ |
| VIFO-eb | $1.544 \pm 0.004$ | $1.724 \pm 0.045$ | $1.895 \pm 0.054$ | $1.785 \pm 0.028$ |
| VI-naive | $1.222 \pm 0.013$ | | | |
| VI-mean | $1.174 \pm 0.016$ | | | |
| VI-mv | $1.078 \pm 0.014$ | | | |
| VI-eb | $1.219 \pm 0.024$ | | | |
| SGD | $1.375 \pm 0.007$ | | | |
| SWA | $1.389 \pm 0.011$ | | | |
| SWAG | $1.464 \pm 0.005$ | | | |
| Repulsive | $2.014 \pm 0.011$ | | | |
| Dir | $2.252 \pm 0.002$ | | | |
| Dropout | $1.053 \pm 0.014$ | | | |
| Laplace | $1.506 \pm 0.008$ | | | |
| VBLL | $1.319 \pm 0.006$ | | | |

Table F.29: AUROC:CIFAR10-SVHN, AlexNet

| method | $\eta_{\text{aux}} = 0$ | $\eta_{\text{aux}} = 0.1$ | $\eta_{\text{aux}} = 0.5$ | $\eta_{\text{aux}} = 1.0$ |
|---|---|---|---|---|
| VIFO-naive | $0.860 \pm 0.005$ | $0.857 \pm 0.008$ | $0.833 \pm 0.018$ | $0.853 \pm 0.018$ |
| VIFO-mean | $0.873 \pm 0.005$ | $0.855 \pm 0.008$ | $0.858 \pm 0.007$ | $0.856 \pm 0.008$ |
| VIFO-mv | $0.893 \pm 0.007$ | $0.871 \pm 0.004$ | $0.873 \pm 0.004$ | $0.861 \pm 0.006$ |
| VIFO-eb | $0.860 \pm 0.007$ | $0.859 \pm 0.007$ | $0.864 \pm 0.002$ | $0.844 \pm 0.010$ |
| VI-naive | $0.898 \pm 0.016$ | | | |
| VI-mean | $0.886 \pm 0.029$ | | | |
| VI-mv | $0.893 \pm 0.009$ | | | |
| VI-eb | $0.885 \pm 0.010$ | | | |
| SGD | $0.851 \pm 0.004$ | | | |
| SWA | $0.862 \pm 0.009$ | | | |
| SWAG | $0.863 \pm 0.005$ | | | |
| Repulsive | $0.857 \pm 0.020$ | | | |
| Dir | $0.846 \pm 0.006$ | | | |
| Dropout | $0.822 \pm 0.011$ | | | |
| Laplace | $0.871 \pm 0.006$ | | | |
| VBLL | $0.889 \pm 0.003$ | | | |

Table F.30: AUROC:STL10-SVHN, AlexNet

| method | $\eta_{\text{aux}} = 0$ | $\eta_{\text{aux}} = 0.1$ | $\eta_{\text{aux}} = 0.5$ | $\eta_{\text{aux}} = 1.0$ |
|---|---|---|---|---|
| VIFO-naive | $0.787 \pm 0.005$ | $0.776 \pm 0.009$ | $0.781 \pm 0.010$ | $0.755 \pm 0.015$ |
| VIFO-mean | $0.789 \pm 0.010$ | $0.757 \pm 0.006$ | $0.768 \pm 0.013$ | $0.774 \pm 0.006$ |
| VIFO-mv | $0.798 \pm 0.008$ | $0.772 \pm 0.010$ | $0.764 \pm 0.016$ | $0.779 \pm 0.013$ |
| VIFO-eb | $0.793 \pm 0.007$ | $0.793 \pm 0.010$ | $0.769 \pm 0.009$ | $0.758 \pm 0.005$ |
| VI-naive | $0.818 \pm 0.020$ | | | |
| VI-mean | $0.792 \pm 0.011$ | | | |
| VI-mv | $0.775 \pm 0.018$ | | | |
| VI-eb | $0.736 \pm 0.044$ | | | |
| SGD | $0.750 \pm 0.010$ | | | |
| SWA | $0.761 \pm 0.010$ | | | |
| SWAG | $0.769 \pm 0.005$ | | | |
| Repulsive | $0.799 \pm 0.013$ | | | |
| Dir | $0.779 \pm 0.004$ | | | |
| Dropout | $0.681 \pm 0.017$ | | | |
| Laplace | $0.783 \pm 0.006$ | | | |
| VBLL | $0.783 \pm 0.015$ | | | |

Table F.31: AUROC:SVHN-CIFAR10, AlexNet

| method | $\eta_{\text{aux}} = 0$ | $\eta_{\text{aux}} = 0.1$ | $\eta_{\text{aux}} = 0.5$ | $\eta_{\text{aux}} = 1.0$ |
|---|---|---|---|---|
| VIFO-naive | $0.969 \pm 0.001$ | $0.968 \pm 0.001$ | $0.967 \pm 0.001$ | $0.967 \pm 0.001$ |
| VIFO-mean | $0.976 \pm 0.001$ | $0.974 \pm 0.001$ | $0.973 \pm 0.001$ | $0.973 \pm 0.002$ |
| VIFO-mv | $0.972 \pm 0.001$ | $0.973 \pm 0.001$ | $0.972 \pm 0.000$ | $0.973 \pm 0.001$ |
| VIFO-eb | $0.969 \pm 0.001$ | $0.971 \pm 0.001$ | $0.970 \pm 0.001$ | $0.973 \pm 0.003$ |
| VI-naive | $0.970 \pm 0.002$ | | | |
| VI-mean | $0.963 \pm 0.014$ | | | |
| VI-mv | $0.965 \pm 0.003$ | | | |
| VI-eb | $0.967 \pm 0.003$ | | | |
| SGD | $0.962 \pm 0.001$ | | | |
| SWA | $0.963 \pm 0.001$ | | | |
| SWAG | $0.968 \pm 0.001$ | | | |
| Repulsive | $0.972 \pm 0.002$ | | | |
| Dir | $0.976 \pm 0.001$ | | | |
| Dropout | $0.952 \pm 0.003$ | | | |
| Laplace | $0.971 \pm 0.002$ | | | |
| VBLL | $0.970 \pm 0.001$ | | | |

Table F.32: AUROC:SVHN-STL10, AlexNet

| method | $\eta_{\text{aux}} = 0$ | $\eta_{\text{aux}} = 0.1$ | $\eta_{\text{aux}} = 0.5$ | $\eta_{\text{aux}} = 1.0$ |
|---|---|---|---|---|
| VIFO-naive | $0.971 \pm 0.000$ | $0.972 \pm 0.001$ | $0.969 \pm 0.001$ | $0.971 \pm 0.001$ |
| VIFO-mean | $0.976 \pm 0.000$ | $0.976 \pm 0.000$ | $0.977 \pm 0.001$ | $0.977 \pm 0.001$ |
| VIFO-mv | $0.976 \pm 0.001$ | $0.975 \pm 0.000$ | $0.977 \pm 0.001$ | $0.976 \pm 0.000$ |
| VIFO-eb | $0.974 \pm 0.001$ | $0.974 \pm 0.001$ | $0.973 \pm 0.001$ | $0.977 \pm 0.002$ |
| VI-naive | $0.974 \pm 0.003$ | | | |
| VI-mean | $0.968 \pm 0.013$ | | | |
| VI-mv | $0.970 \pm 0.002$ | | | |
| VI-eb | $0.971 \pm 0.002$ | | | |
| SGD | $0.966 \pm 0.001$ | | | |
| SWA | $0.968 \pm 0.002$ | | | |
| SWAG | $0.972 \pm 0.000$ | | | |
| Repulsive | $0.975 \pm 0.002$ | | | |
| Dir | $0.979 \pm 0.001$ | | | |
| Dropout | $0.958 \pm 0.004$ | | | |
| Laplace | $0.975 \pm 0.002$ | | | |
| VBLL | $0.973 \pm 0.000$ | | | |

Table F.33: AUROC:CIFAR10-SVHN, PreResNet20

| method | $\eta_{\text{aux}} = 0$ | $\eta_{\text{aux}} = 0.1$ | $\eta_{\text{aux}} = 0.5$ | $\eta_{\text{aux}} = 1.0$ |
|---|---|---|---|---|
| VIFO-naive | $0.885 \pm 0.004$ | $0.880 \pm 0.006$ | $0.889 \pm 0.005$ | $0.898 \pm 0.006$ |
| VIFO-mean | $0.894 \pm 0.004$ | $0.889 \pm 0.008$ | $0.907 \pm 0.007$ | $0.896 \pm 0.002$ |
| VIFO-mv | $0.914 \pm 0.002$ | $0.919 \pm 0.007$ | $0.917 \pm 0.001$ | $0.915 \pm 0.003$ |
| VIFO-eb | $0.908 \pm 0.002$ | $0.912 \pm 0.005$ | $0.918 \pm 0.002$ | $0.913 \pm 0.004$ |
| VI-naive | $0.863 \pm 0.008$ | | | |
| VI-mean | $0.868 \pm 0.011$ | | | |
| VI-mv | $0.868 \pm 0.008$ | | | |
| VI-eb | $0.858 \pm 0.013$ | | | |
| SGD | $0.950 \pm 0.001$ | | | |
| SWA | $0.950 \pm 0.001$ | | | |
| SWAG | $0.963 \pm 0.001$ | | | |
| Repulsive | $0.819 \pm 0.004$ | | | |
| Dir | $0.962 \pm 0.002$ | | | |
| Dropout | $0.837 \pm 0.010$ | | | |
| Laplace | $0.957 \pm 0.002$ | | | |
| VBLL | $0.902 \pm 0.003$ | | | |

Table F.34: AUROC:STL10-SVHN, PreResNet20

| method | $\eta_{\text{aux}} = 0$ | $\eta_{\text{aux}} = 0.1$ | $\eta_{\text{aux}} = 0.5$ | $\eta_{\text{aux}} = 1.0$ |
|---|---|---|---|---|
| VIFO-naive | $0.810 \pm 0.005$ | $0.810 \pm 0.004$ | $0.804 \pm 0.004$ | $0.803 \pm 0.003$ |
| VIFO-mean | $0.810 \pm 0.001$ | $0.835 \pm 0.006$ | $0.836 \pm 0.004$ | $0.820 \pm 0.003$ |
| VIFO-mv | $0.788 \pm 0.004$ | $0.794 \pm 0.001$ | $0.789 \pm 0.003$ | $0.784 \pm 0.005$ |
| VIFO-eb | $0.799 \pm 0.004$ | $0.800 \pm 0.005$ | $0.803 \pm 0.002$ | $0.803 \pm 0.004$ |
| VI-naive | $0.788 \pm 0.014$ | | | |
| VI-mean | $0.784 \pm 0.006$ | | | |
| VI-mv | $0.788 \pm 0.015$ | | | |
| VI-eb | $0.728 \pm 0.030$ | | | |
| SGD | $0.795 \pm 0.008$ | | | |
| SWA | $0.837 \pm 0.008$ | | | |
| SWAG | $0.877 \pm 0.005$ | | | |
| Repulsive | $0.804 \pm 0.005$ | | | |
| Dir | $0.812 \pm 0.004$ | | | |
| Dropout | $0.689 \pm 0.007$ | | | |
| Laplace | $0.810 \pm 0.004$ | | | |
| VBLL | $0.810 \pm 0.008$ | | | |

Table F.35: AUROC:SVHN-CIFAR10, PreResNet20

| method | $\eta_{\mathrm{aux}} = 0$ | $\eta_{\mathrm{aux}} = 0.1$ | $\eta_{\mathrm{aux}} = 0.5$ | $\eta_{\mathrm{aux}} = 1.0$ |
|---|---|---|---|---|
| VIFO-naive | $0.890 \pm 0.005$ | $0.958 \pm 0.003$ | $0.934 \pm 0.010$ | $0.948 \pm 0.004$ |
| VIFO-mean | $0.893 \pm 0.002$ | $0.931 \pm 0.010$ | $0.925 \pm 0.006$ | $0.928 \pm 0.014$ |
| VIFO-mv | $0.912 \pm 0.004$ | $0.936 \pm 0.005$ | $0.927 \pm 0.007$ | $0.916 \pm 0.006$ |
| VIFO-eb | $0.916 \pm 0.007$ | $0.922 \pm 0.002$ | $0.923 \pm 0.008$ | $0.941 \pm 0.009$ |
| VI-naive | $0.870 \pm 0.008$ | | | |
| VI-mean | $0.867 \pm 0.016$ | | | |
| VI-mv | $0.866 \pm 0.012$ | | | |
| VI-eb | $0.847 \pm 0.026$ | | | |
| SGD | $0.909 \pm 0.006$ | | | |
| SWA | $0.918 \pm 0.005$ | | | |
| SWAG | $0.919 \pm 0.006$ | | | |
| Repulsive | $0.843 \pm 0.006$ | | | |
| Dir | $0.970 \pm 0.003$ | | | |
| Dropout | $0.830 \pm 0.028$ | | | |
| Laplace | $0.890 \pm 0.006$ | | | |
| VBLL | $0.906 \pm 0.004$ | | | |

Table F.36: AUROC:SVHN-STL10, PreResNet20

| method | $\eta_{\mathrm{aux}} = 0$ | $\eta_{\mathrm{aux}} = 0.1$ | $\eta_{\mathrm{aux}} = 0.5$ | $\eta_{\mathrm{aux}} = 1.0$ |
|---|---|---|---|---|
| VIFO-naive | $0.894 \pm 0.009$ | $0.957 \pm 0.009$ | $0.934 \pm 0.008$ | $0.936 \pm 0.010$ |
| VIFO-mean | $0.896 \pm 0.004$ | $0.924 \pm 0.013$ | $0.914 \pm 0.010$ | $0.921 \pm 0.009$ |
| VIFO-mv | $0.917 \pm 0.006$ | $0.933 \pm 0.008$ | $0.928 \pm 0.009$ | $0.922 \pm 0.001$ |
| VIFO-eb | $0.911 \pm 0.006$ | $0.916 \pm 0.005$ | $0.923 \pm 0.004$ | $0.936 \pm 0.006$ |
| VI-naive | $0.870 \pm 0.010$ | | | |
| VI-mean | $0.867 \pm 0.016$ | | | |
| VI-mv | $0.865 \pm 0.011$ | | | |
| VI-eb | $0.846 \pm 0.027$ | | | |
| SGD | $0.907 \pm 0.007$ | | | |
| SWA | $0.913 \pm 0.008$ | | | |
| SWAG | $0.915 \pm 0.007$ | | | |
| Repulsive | $0.847 \pm 0.007$ | | | |
| Dir | $0.970 \pm 0.003$ | | | |
| Dropout | $0.829 \pm 0.029$ | | | |
| Laplace | $0.911 \pm 0.007$ | | | |
| VBLL | $0.909 \pm 0.003$ | | | |

