# OpenReview forum: "Variational Inference on the Final-Layer Output of Neural Networks"
_TMLR — Accepted by TMLR_

### Review · Reviewer_wCD2 · 2024-06-16

**Summary Of Contributions:**

This paper proposes applying variational inference (VI) to not the parameter space of neural networks but solely the final layer output space. This approach, called VIFO, aims at estimating the mean and variance of the output with a reduced computational cost compared to the standard VI. The theoretical analysis of VIFO's representational power demonstrates that it is equivalent to that of VI for linear models, while strictly less powerful for non-linear models. The efficiency and effectiveness of the proposed VIFO method have been empirically validated using real-world datasets.

**Audience:**

Yes

**Broader Impact Concerns:**

I do not have any concerns.

**Claims And Evidence:**

Yes

**Requested Changes:**

Please address the concerns outlined in the Weaknesses section above.

**Strengths And Weaknesses:**

### Strengths
- This paper is generally well-written, and the presentation is clear.
- The properties of the proposal have been carefully examined both theoretically and empirically, ensuring high overall quality. Notably, the representation power (expressiveness) of the proposed VIFO has been theoretically analyzed, presenting both positive (the linear case) and negative (the non-linear case) results.
- The novelty of the proposal is sufficient for a TMLR paper, in my opinion.
### Weaknesses
- Since one of the advantages of VIFO is its lower computational cost, the time complexity of VIFO and its comparison to VI should be presented.
- In the analysis of expressiveness in Section 3, the dimensionality of $\theta$ is assumed to be the same as that of an input vector, $d$. However, this assumption does not hold in typical situations, as neural networks tend to have a much larger number of parameters. How does this assumption impact the theoretical results (Theorems 3.1, 3.2)?
- In the experiments, the runtime for training 1 epoch has been measured. However, since the number of epochs can vary among compared methods, it would be valuable to show and compare the total runtime for training.

---

> ### Author Response · Authors · 2024-06-24
>
> We appreciate your detailed review and constructive feedback on our manuscript. Below, we address your concerns:
>
> - *Time complexity:*
> Let $P$ denote the number of parameters in the **base** model and $M$ the number of samples used for Bayesian model averaging. We mainly consider two parts: the forward/backward pass and the loss computation. Each forward pass takes $O(P)$ time. Using reverse mode automatic differentiation the backward pass also takes $O(P)$ time, and the same holds for updates. The time complexity for computing the loss for each output of the base model is $O(1)$.
> For VI, we need to sample $M$ sets of parameters. Thus, for prediction we perform $M$ forward passes and for learning we perform $M$ forward/backward passes and compute the loss $M$ times. Therefore, the time complexity for VI is $O(PM + M) = O(PM)$.
> For VIFO, we perform one forward/backward pass and sample $M$ times in the output space. This means we compute the loss $M$ times, resulting in a time complexity for VIFO of $O(P + M)$.
> For Stochastic Gradient Descent (SGD), we perform a forward/backward pass once without any sampling and compute the loss once, giving a time complexity of $O(P)$.
> Typically, $P \gg M$, so the additional time complexity introduced by VIFO compared with SGD is negligible.
>
> - *Dimension of $\theta$:*
> The assumption that $\theta$ has the same dimension as the input vector is necessary only for generalized linear regression case, and thus it is required for the positive result of Theorem 3.1. However, Theorem 3.2 applies to all non-linear models and does not rely on this assumption. Violating this assumption does not invalidate Theorem 3.2.
>
> - *Epochs:*
> In our experiments, we set the number of epochs to 500 for all methods. This makes for an easy comparison because otherwise the time to "convergence" depends on the convergence criterion chosen which may need to be different for different methods.
> Having said that, looking at learning curves,
> we observe that VIFO converges as quickly as, or faster than, VI, resulting in a shorter total run time for VIFO compared to VI in terms of test loss. We include learning curve plots in the revised version of our paper to clarify this point. Please refer to section E.4. in the revised version.
>
> Thank you for your valuable input. Please let us know if you have any further suggestions or need additional clarifications.
>
> Sincerely,
>
> Authors

---

### Review · Reviewer_Q3xj · 2024-06-18

**Summary Of Contributions:**

This paper describes variational inference applied to the model final-layer outputs (VIFO), directly predicting the mean and variance of each output unit (logits for classification). The authors optimize an evidence lower bound (ELBO), incorporating auxiliary training and collapsed variational inference. VIFO has the same expressiveness as standard variational inference for linear models, but is less powerful for deep networks. In practice, as VIFO may be trained fairly efficiently, this may be mitigated by using ensembles of VIFO models. On a few computer vision datasets, VIFO achieves similar (and sometimes superior) in-distribution performance as both deterministic and bayesian baselines, and also captures uncertainty in out-of-distribution data.

**Audience:**

Yes

**Claims And Evidence:**

Yes

**Requested Changes:**

1. You mention that $W_1$ and $W_2$ have shared components. What exactly is shared, and what isn't?

2. In Eq. 2, as g(l) as the normal distribution variance, should the notation be $\mathcal{N}(y|m, g(l))$ instead of $\mathcal{N}(y|m, l)$?

3. Could you briefly describe similarities and differences between your proposed approach and [1], which was very recently published?

4. Could you elaborate on the relationship between your work and the local reparametrization trick, which also moves the noise from the model weights to its activations (although not necessarily the last layer only)?

5. Before introducing your approach, it could be helpful to briefly recap how standard (and deterministic) variational inference works.

6. Table 1: mention time unit.

7. How many models are there in each ensemble?

[1] Harrison et al. Variational Bayesian Last Layers. ICLR 2024. https://arxiv.org/pdf/2404.11599

**Strengths And Weaknesses:**

**Strengths**

The model is reasonably simple and appears to achieve a good tradeoff between efficiency, in-distribution performance and out-of-distribution uncertainty quantification.

The paper provides both empirical evidence and theoretical justifications.

**Weaknesses**

The abstract states that "we incorporate collapsed variational inference with VIFO which significantly improves the performance in practice". However, this is not clearly established by experiments in the main paper (although there are additional results in the appendix). Likewise, it isn't clear how performance scales with the number of models in the ensemble.

---

> ### Author Response · Authors · 2024-06-24
>
> Thank you for your insightful review and valuable feedback on our manuscript. We have revised the paper, fixing the typos and adding additional discussion. Below are our detailed responses to your concerns:
>
> - *Size of Ensembles and Effect of Collapsed Variational Inference:*
> We used 5 models in the ensembles in all experiments in the submission. To examine how the number of ensembles affects performance, we have included figures in the revised version of our paper. Please refer to Section E.5. These figures show that, as expected, increasing the size of the ensemble further improves the performance but that the effect becomes smaller with increasing size. The curves include all variants of VIFO and demonstrate that collapsed VIFO improves performance over the basic variant.
>
> - *Shared Components:*
> The key aspect of VIFO is that it requires two outputs: one for the mean and the other for the variance. We can achieve this by using two independent neural networks, one for each output, where
> $W_1$ and $W_2$ do not share any components. Alternatively, we can use a single neural network with twice the number of final outputs, with the first half representing the mean and the second half representing the variance. In this setup, which was used in the experiments in the paper, all parameters are shared except for those in the final layer.
>
> - *Eq 2:* Yes. It is a typo and we will fix this.
>
> - *Comparison to VBLL[1]:*
> VBLL performs variational inference on the last layer similar to (Brosse et al., 2020) which is discussed in section 5 in our paper.
> VBLL improves on prior work by replacing the sampling or integration which is needed for VI at the last layer with an approximate analytic computation which incorporates log loss. Hence VBLL improves the computational complexity of last layer models but does not change the model itself.
> As a result, VBLL and VIFO differ in how they regularize training and in the effective computation of the loss. For VBLL, as in VI, a single KL regularization is for all inputs, whereas for VIFO, the KL regularization term varies with each input. As seen in Equation 3, the KL regularization between $q(z|x)$ and $p(z|x)$ depends on the input $x$. Secondly, the methods differ in how they compute the loss function: we use reparametrization, while VBLL employs an approximate analytical solution.
>
> - *Comparison to Local Reparametrization Trick (LRT):*
> For LRT, let $W_{1, l}, W_{2, l}$ denote the mean and variance of weights at layer $l$ respectively.
> For each layer, $\mu_l = f(W_{1, l}, x_l)$ and $\sigma_l = f(W_{2, l}, x_l^2)$, where $f$ is either a linear function or a convolutional function. We then sample and pass the result to the activation function: $x_l = \text{act}(\mu_l + \epsilon \sigma_l)$. This procedure is repeated for each layer until the final output $x_L$ is obtained. In contrast, VIFO does not involve sampling in these intermediate layers. Instead, we directly compute $\mu_L = f_{W_1}(x)$ and $\sigma_L = f_{W_2}(x)$, and then sample $x_L = \mu_L + \epsilon \sigma_L$. From this, we can see that VIFO is more efficient.
>
> Please let us know if you have further suggestions or require additional clarifications.
>
> Best,
>
> Authors

---

### Review · Reviewer_c4nX · 2024-06-30

**Summary Of Contributions:**

The authors introduce the VIFO method, including parameterization of the model, introduction of auxiliary losses, and discussion of collapsed VI methods. The variational formulation is similar to variational policy parameterization in reinforcement learning and VAEs, in which the KL divergence of the function output is regularized, as opposed to model parameters (as in Bayesian NNs).

**Audience:**

Yes

**Broader Impact Concerns:**

No ethical concerns.

**Claims And Evidence:**

Yes

**Requested Changes:**

The following changes should be made:
- Strengthen baselines, as described above (single-pass uncertainty methods, standard ensembles)
- Add a discussion of similarities to VIFO
- Discuss the novelty of the collapsed VI results.
- Fix typos.

**Strengths And Weaknesses:**

Strengths:
- The method is reasonably simple (for a Bayesian method) and computationally efficient.
- The discussion of representational capacity of the model is a good contribution, but should be expanded upon (discussed below).

Weaknesses:
- From figure 1, the method without auxiliary training seems very weak. The introduction of the auxiliary loss is reasonable, but the discussion of this term is lacking and seems essential for the method.
  - The authors _must_ discuss how ablations of the sampling distribution for x_aux, how eta_aux should be tuned in practice, and any other details.
- The baselines compared to are not sufficient. There are many comparable inexpensive single-pass uncertainty quantification methods that should be compared to
  - for example: DUQ, SNGP, VBLL, last layer Laplace (these methods should also all be discussed in the related work)
- VIFO is very similar to deep VIBs (Alemi et al, 2017), and that work should be discussed.
- As far as I can tell, much of the presentation of the collapsed VI (including in appendix C) is very similar to Tomczak et al (2021). The authors should precisely lay out what is novel in their presentation.
- While section 3 discusses the ability of VIFO to match the output of a linear variational model, it doesn't discuss the difference in regularization terms between these two models, which may be interested for understanding the output-VI approach.
- The Rademacher complexity analysis is not particularly important to the paper and, while it is a reasonable technical result, does not add much to the paper overall. It could be moved to the appendix, ideally to make room for a more robust experimental characterization.
- The results presented are for ensembles of VIFO. This is highly misleading, and the authors should:
  - Include results for a single VIFO model
  - Make sure to label VIFO ensembles as VIFO ensembles
  - Include an ensemble of standard models as a baseline

Small comments:
- Several typos: "Beyesian" in the abstract, VIFO not consistently capitalized
- The discussion of the model in section 2 is a bit unclear. For example, the presentation of the regression model (having four outputs) is unclear and should be more formally presented. In general, I would recommend the authors make one clear subsection on classification and one on regression.
- The authors repeatedly state that they perform variational inference on the _output layer_, which is misleading. The authors perform VI on the _outputs_, and the output layer (corresponding to the weights) are not regularized (by a term KL penalty).
- The presentation of figures 3-6 probably do not need to be plots, and could be tables instead.
- Inclusion of arrows for "which direction is better" in metrics is useful for casual readers; currently the authors include it for some figures but it would be helpful to include for all.

---

> ### Author Response · Authors · 2024-07-11
>
> Thank you for your thorough review of our manuscript and for your constructive feedback. We greatly appreciate the time and effort you have invested in providing your comments. We have revised the paper accordingly. Below are our responses to your concerns:
>
> - *Auxiliary training:* In this paper we use only a uniform sampling distribution for $x_\text{aux}$, as is also done in functional variational inference [1] and last-layer Laplace [2]. We leave the exploration of other sampling distributions for future work and believe that better sampling distributions could enhance VIFO's performance. For $\eta_{\text{aux}}$, we experiment with $\eta_{\text{aux}} \in$ {0.0, 0.1, 0.5, 1.0}. Larger values of $\eta_{\text{aux}}$ generally improve OOD data detection but at the cost of increased in-distribution loss, and there is no generic optimal value $\eta_{\text{aux}}$.
> We include results for all values of $\eta_{\text{aux}}$ in the appendix and present only the case when $\eta_{\text{aux}}$ is 0.1 in the main text, as it provides a balance between in-distribution log loss and OOD performance.
>
> - *Single and Ensembles:*
> Please note that, as mentioned in section 6.2,
> in our experiments, all methods except VI and repulsive ensembles are compared using ensembles of five single models.  In particular, the ensembles of standard models are labeled as SGD in our plots and are already included in our paper.
> For this reason we did not explicitly label VIFO as VIFO ensembles (we would need to label all methods in this manner).
> We have included additional results for a single VIFO model in Appendix E.5, which show how performance changes with increasing ensemble size.
>
> - *New baselines and related work:* Please see detailed discussion of these methods in the updated related works section. We have chosen to add (ensembles of) VBLL and last-layer Laplace (LL Laplace) in the new revision for comparisons because they are more closely related to VIFO in this group, and because VBLL is the most recently published of these.
> VBLL and LL Laplace generally have worse log loss and accuracy than VIFO in most cases, and worse ECE and entropy in all cases.
>
> - *Deep VIB:* Please see detailed discussion in the updated related works section. Deep VIB is motivated from a different perspective but the equation of its final optimization objective is similar to VIFO. However, in their formulation, $z$ is the output of a bottleneck layer which is not the final layer (because it is meant to constrain the information that flows to the final layer), and $p(z)$ which is the prior in our model is a posterior on the marginal posterior on $z$. Nonetheless, our development of rich priors through collapsed inference can help inform the choice of $p(z)$ in that model, which is typically taken to be a standard Normal.
>
> - *Collapsed VI:* Our contribution is the application of collapsed VI to VIFO (i.e., to priors on $z$ instead of $W$) and the derivation of empirical Bayes as a special case of collapsed VI. We mention in the paper that the derivations follow the same methodology and steps as Tomczak et al. (2021) but the variables and equations are different. We have therefore included them in the appendix rather than in the main paper.
>
> - *Regularization in linear models:* VIFO regularization is identical to VI regularization if the prior is data-dependent. However, with a data-independent prior, VIFO cannot recover the regularization term, which may limit its expressiveness.
>
> Once again, thank you for your valuable insights. Please let us know if you have any further suggestions or require additional clarifications.
>
> Best regards,
>
> Authors

---

### Decision · Action_Editor_3cW4 · 2024-09-11

**Recommendation:** Accept as is

**Comment:**

This paper develops a method for marginalizing over the final layer of a deep neural network through using variational inference.  Such "last-layer" strategies are quite appealing, as they can be done far more efficiently than integrating over all the parameters of the model while still retaining some of the benefits of a Bayesian approach.  The reviewers all found that the paper was well written, interesting, sufficiently novel and relevant to the TMLR audience.  Initially multiple reviewers found the experiments lacked comparison to state-of-the-art baselines.  However, the authors added significant additional experiments, particularly to the highly related VBLL method, and the reviewers seemed satisfied with this new comparison.   Although the work is novel enough for TMLR, the reviewers pointed out that there is a significant amount of quite related work that should be discussed (and is in the latest version).  One reviewer suggested moving the Rademacher complexity section to the appendix (optional for the camera ready).

In the field, researchers have drawn out a kind of Pareto frontier of computational / memory efficiency of uncertainty quantification vs quality of the uncertainty estimates.  This paper provides a nice addition to the highly efficient part of that curve.  Overall a nice paper that provides a practically useful contribution.  The reviewers all voted to accept, so the recommendation is to accept the paper as is.  Please include the changes from the last revision in the camera ready version.

**Audience:**

Efficient uncertainty quantification for deep learning is an active area of research in ML and practically important for model reliability.

**Claims And Evidence:**

The reviewers all found that the claims were backed by sufficient empirical and theoretical evidence.  In particular, the variational method proposed by the authors is compared to relevant baselines and is shown to be on par or sometimes better, with often significant computational savings.